# INTERACTION BETWEEN ELEVATED CO₂ AND PHYTOPLANKTON-DERIVED ORGANIC MATTER UNDER SOLAR RADIATION ON BACTERIAL METABOLISM FROM COASTAL WATERS

Antonio Fuentes-Lema[1]; Henar Sanleón-Bartolomé[2]; Luis M. Lubián[3]; Cristina Sobrino[1*]

[1] - UVigo Marine Research Centre; Lagoas Marcosende Campus 36331 Vigo, Spain

[2] - Spanish Institute of Oceanography (IEO). Paseo Marítimo Alcalde Francisco Vázquez 10,

15001 A Coruña, Spain.

[3] - Institute of Marine Sciences of Andalucía (CSIC). Campus Univ. Rio San Pedro. 11519 Puerto Real,

Cádiz, Spain

*Correspondence to: Cristina Sobrino. +34 986 818789, sobrinoc@uvigo.es

       Running head: Ocean acidification and DOM on bacteria

**Abstract.** Microcosm experiments to assess bacterioplankton response to phytoplankton-derived organic

matter obtained under current and future-ocean CO₂ levels were performed. Surface seawater enriched

with inorganic nutrients was bubbled for 8 days with air (current CO₂ scenario) or with a 1000 ppm CO₂–

air mixture (future CO₂ scenario) under solar radiation. The organic matter produced under the current

and future CO₂ scenarios was subsequently used as inoculum. Triplicate 12 L flasks filled with 1.2 µm-

filtered natural seawater enriched with the organic matter inocula were incubated in the dark for 8 days

under $CO_2$ conditions simulating current and future $CO_2$ scenarios to study the bacterial response. The acidification of the media increased bacterial respiration at the beginning of the experiment while the addition of the organic matter produced under future levels of $CO_2$ was related to changes in bacterial

production and abundance. The balance between both, respiration and production, made that the bacteria grown under future $CO_2$ levels with an addition of non-acidified matter showed the best growth efficiency at the end of the incubation. However cells grown under future scenarios with high $CO_2$ levels and acidified organic matter additions did not perform differently than those grown under present $CO_2$ conditions, independently of the addition of acidified or non-acidified organic matter. This study

demonstrates that the increase in atmospheric $CO_2$ concentrations can affect bacterioplankton directly by changes in the respiration rate and indirectly by changes on the organic matter with concomitant effects on bacterial production and abundance.

KEY WORDS: bacterioplankton, phytoplankton, organic matter, ocean acidification.



# 1 Introduction:

The increase in fossil fuel burning, cement production and deforestation together with changes in land use have resulted in an accumulation of atmospheric $CO_2$ at levels never seen before the last two million years (Caldeira & Wickettt 2008, Le Quere et al. 2015). Atmospheric gases can freely diffuse into the ocean surface, which has already absorbed about 30% of the emitted anthropogenic $CO_2$, perturbing the carbonate system and decreasing ocean pH in a process known as ocean acidification (Sabine et al. 2004, Burke et al. 2014).

The latest IPCC report shows that the pH of surface ocean waters has already decreased by 0.1, corresponding to a 26% increase in acidity. If mitigation strategies for global change are not adopted and $CO_2$ emissions continue as usual, ocean pH values will drop about 0.3-0.7 units by the end of the $21^{st}$ century (Burke et al. 2014). The decrease in seawater pH has strong effects on the ecosystem, the aquatic organisms and the interactions among them. Studies about its consequences in the surface ocean have been primarily focused on calcifying organisms such as corals or coralline algae because they participate in the formation of habitats and human services (Langdon et al. 2003, Fabry et al. 2008). Recent meta-analysis studies also revealed decreased survival, growth, development and abundance of a broad range of marine organisms, although the magnitude of these responses varies between taxonomic groups, including variation within similar species (Kroeker et al. 2013). Additionally, other authors have demonstrated that ocean acidification can increase growth, primary production and $N_2$ fixation rates in some phytoplankton species (Barcelos e Ramos et al. 2007, Fu et al. 2007, Levitan et al. 2007, Iglesias-Rodriguez et al. 2008). In contrast with the abundant information about phytoplankton, very little is known about the response of their heterotrophic counterparts.

Heterotrophic bacteria play an important role in the planktonic community since they are responsible for the majority of the organic matter remineralization (Cole et al. 1988, Azam et al. 1998, Nagata et al. 2000) allowing the primary producers to make use of the recycled inorganic nutrients. They also return dissolved organic carbon (DOC) to the marine food web via its incorporation into bacterial biomass through what
it is called the microbial loop (Azam et al. 1983). The other way around, the microbial response can change depending on phytoplankton taxonomic composition and the nutrient levels, and therefore productivity, of the water (Teira et al. 2012, Bunse et al. 2016, Sala et al. 2016). Despite this important role of bacterioplankton in the marine food web and biogeochemical cycles, only few studies have been designed to elucidate the effects of ocean acidification on bacteria metabolism or its interaction with the
abiotic or biotic factors. Some of these studies suggest an absence of significant metabolic responses in experiments where $CO_2$ levels were manipulated (Rochelle-Newall et al. 2004, Allgaier et al. 2008, Newbold et al. 2012). Adaptation towards a fast acclimation to low pH values might have occurred since occasionally bacteria and other heterotrophic organisms, suffer lower pH values than those predicted by ocean acidification scenarios (Joint et al. 2011). On the contrary, other authors have reported that a
decrease in seawater pH can potentially influence bacterial metabolism by changes in bacterial production and growth rates in natural communities, although the results show different responses depending on the study (Coffin et al. 2004, Grossart et al. 2006, Motegi et al. 2013, Spilling et al. 2016). More recently, results from a phytoplankton bloom mesocosm study have demonstrated, through metatranscriptome analysis, that acidification can enhance the expression of genes encoding proton pumps to maintain
homeostasis under high $CO_2$ conditions (Bunse et al. 2016).

An interesting point is that the experimental design in most of the published $CO_2$ studies did not allow

distinguishing between direct effects on the bacteria *per se* and indirect effects, due for example to

changes in phytoplankton community composition or to changes in organic matter. Therefore, indirect

pathways such as those affecting the availability of organic matter in terms of quantity, because of an

increase in phytoplankton primary production (Hein & Sand-Jensen 1997, Riebesell et al. 2007), or in

terms of quality, because of changes in the composition of phytoplankton-derived organic matter (Engel

et al. 2014), should be also studied to determine the effect of ocean acidification on bacterioplankton. For

example, recent results have demonstrated that pH values predicted for future ocean acidification

scenarios were close to the optimum values for extracellular enzyme activity (Grossart et al. 2006, Piontek

et al. 2010, Yamada & Suzumura 2010). Higher enzymatic activity resulted in a higher rate of organic

matter transformation and increases in bacteria performance. It has been also demonstrated that decreases

in pH might increase the rate of extracellular dissolved organic carbon production from phytoplankton

(DOCp) and the formation of transparent exopolymer particles (TEPs) (Engel 2002, Engel et al. 2004),

although the contrary has also been observed and there is not a clear response to this matter (Sobrino et

al. 2014). In addition, a higher $CO_2$ concentration in seawater could modify the C:N:P ratios of particulate

organic matter, which may substantially affect the activity of bacteria and the carbon fluxes in the future

ocean (Riebesell et al. 2007, Engel et al. 2014).

Environmental drivers such as the ultraviolet radiation (UV: 280-400 nm) should be also taken into

account when studying the biological and chemical responses of the planktonic assemblages and the

environment, since it plays a crucial role on the physiology of plankton communities and on the ecology

of the aquatic ecosystems. UVR induces photomineralization of coloured dissolved organic matter

(CDOM) increasing the biological availability of the resulting DOM (Moran & Zepp 1997). UV radiation can also increase DOCp production in surface waters (Carrillo et al. 2002, Helbling et al. 2013, Fuentes-Lema et al. 2015).

The main goal of our study was to investigate the direct and indirect effects of ocean acidification on the interaction between phytoplankton derived organic matter and bacterial metabolism. We analyzed the changes in bacterial abundance, production, respiration and viability in a coastal plankton community from an upwelling system subjected to current and future $CO_2$ concentrations.

## 2 Materials and Methods:

*2.1 Experimental Setup*

The experiment to determine the response of bacterioplankton communities to phytoplankton-derived organic matter produced under current and future $CO_2$ scenarios were performed in two phases at the Toralla Marine Science Station, from now on ECIMAT (Estación de Ciencias Mariñas de Toralla,
University of Vigo (ecimat.uvigo.es)) (Fig. 1). The first phase consisted in 8 days incubation under full solar radiation (UV radiation included) of natural phytoplankton communities enriched with inorganic nutrients under current and future $CO_2$ conditions. In the second phase, the organic matter obtained from the previous incubation was added to a natural bacterioplankton community to assess the interactions between organic matter amendments and acidification on bacterial metabolism for 8 days (Fig. 2).

Water from 5 m depth was collected with a Niskin bottle on board of the R/V Mytilus from a fixed central station at the Ría de Vigo (42.23 N; Long: 8.78 W. Fig. 1) the 27th of June 2013 and immediately transported to the ECIMAT (approx. 30 min). The Ría de Vigo is a highly productive and dynamic

embayment located in the Northwestern Iberian Peninsula, characterized by the intermittent upwelling of

cold and inorganic nutrient-rich Eastern North Atlantic Central water (Fraga 1981). Upwelled water in

the Iberian system also brings high $CO_2$ concentrations, so annual $pCO_2$ can oscillate between maximum

values of 750 ppm $CO_2$ during the upwelling events and minimum values of 270 ppm $CO_2$ during the

downwelling season (Alvarez et al. 1999, Gago et al. 2003). The sample was pre-screened using a 200

µm sieve to avoid zooplankton grazing and distributed in six, aged and acid washed, UVR transparent 20

L cubitainers (NalgeneR I-Chem Certified Series™ 300 LDPE Cubitainers). The cubitainers were

submerged in two 1500 L tanks located outdoors in an open area free of shadows. The tanks were

connected to a continuous seawater pumping system and covered by a neutral density screen (75%

Transmittance) to assure cooling at the *in situ* temperature and to avoid photoinhibition by the highly

damaging summer irradiances under the static conditions of the cubitainers, respectively. Nitrate,

ammonium and phosphate were added the first and 5th day of experiment to maintain saturating nutrient

conditions (5 µmol L$^{-1}$ nitrate ($NO_3^-$), 5 µmol L$^{-1}$ ammonium ($NH_4^+$) and 1 µmol L$^{-1}$ phosphate ($HPO_4^{-2}$)

final concentrations). Triplicate samples were bubbled with regular atmospheric air (Low Carbon

treatment: LC, aprox. 380 ppm $CO_2$) or with a mixture of the atmospheric air and $CO_2$ from a gas tank

(Air Liquide Spain) (High Carbon treatment: HC, 1000 ppm $CO_2$). At the end of the incubation, the

samples grown under present and future levels of $CO_2$ were stored frozen at -20 ˚C until the start of the

second phase, ten days later, to be used as a naturally-derived organic matter inocula. The inocula included

both, dissolved and particulate organic matter, as observed in nature, but since bacteria preferentially use

the labile dissolved organic matter pool for growth (Nagata et al. 2000, Lechtenfeld et al. 2015) we

focused our measurements of the organic matter on the dissolved fraction.

For the second phase, water collection was similar to the previous one. Once at ECIMAT, seawater was

filtered through 1.2 µm pore size glass fibber filters (GF/C Whatman Filters) to separate the

bacterioplankton community from the other plankton cells, mainly diatoms for this time of the year

(Tilstone et al. 2000, Fuentes-Lema & Sobrino 2010). The bacterioplankton sample was distributed in 12

acid washed polycarbonate NALGENE 12 L bottles together with the phytoplankton derived organic

matter inoculum and 0.2 µm filtered seawater (FSW) following a 3:1:6 bacterioplankton: organic matter:

FSW volume ratio, respectively. This proportion aimed to reach 10 µmol $L^{-1}$ of organic carbon, which

represent around 25-30% of the mean excess of organic carbon observed in the surface layer of the middle

Ría de Vigo as compared with the bottom waters (Doval et al. 1997, Nieto-Cid et al. 2005). Half of the

bottles were inoculated with organic matter produced under current $CO_2$ conditions (Non acidified

Organic matter, NO, n=6) and the other half with organic matter produced under future $CO_2$ conditions

(Acidified Organic matter, AO, n=6). In each case, three replicates were aerated with ambient $CO_2$ levels

(Low Carbon treatment (LC): LC_NO, LC_AO) or air with 1000 ppm $CO_2$ (High Carbon treatment (HC):

HC_NO, HC_AO). This experimental setup produced 4 different treatments from the less modified

sample (LC_NO) to the most altered sample (HC_AO) (Fig. 2). The bottles were located in a walk-in

growth chamber under dark conditions at 15 ºC, similar to *in situ* temperatures, with the aim of focusing

the experiment on the two factors of the study, avoiding potential photoinhibitory and photochemical

effects of solar radiation on bacterioplankton and organic matter, respectively.

*2.2 DIC, pH and $CO_2$ analysis*

Triplicate 30 mL samples were filtered daily through 0.2 μm size pore nitrocellulose filters. The filtrate
was encapsulated without air bubbles in 10 mL serum vials and stored at 4 °C and dark conditions until

analysis immediately after the incubation ended. Dissolved Inorganic Carbon (DIC) analysis was carried

out through acidification with 10% HCl using a $N_2$ bubbler connected to an infrared gas analyzer (LICOR

7000) and calibration was performed using a $Na_2CO_3$ standard curve. pH and temperature were daily

measured with a Crison pH 25 pH meter and salinity with a thermosalinometer Pioneer 30. The pH meter

was calibrated to the total hydrogen ion concentration pH scale with a 2-amino-2-hydroxymethyl-1,3-

propanediol (tris) buffer prepared in synthetic seawater (Dickson & Goyet 1994). The partial pressure of

$CO_2$ ($pCO_2$) in the samples was calculated from salinity, temperature, pH and DIC using the software

csys.m from Zeebe & Wolf-Gladrow (2001).

*2.3 Chlorophyll a concentration*

Seawater samples for Chl *a* analysis were taken every day during the first incubation. A volume of 150

mL from each cubitainer was gently filtered through GF/F filters under dim light and immediately stored

at -20 ˚C until further analysis. For Chl *a* extraction, the filters were kept at 4 °C overnight in acetone

90%. Chl *a* concentration was estimated with a Turner Design Fluorometer TD-700 and a pure Chl *a*

standard solution was used for calibration.

*2.4 Primary production*

Incubations were performed at noon and lasted 3 to 3.5 hours. Fifteen mL samples of each microcosm

were inoculated with $H^{14}CO_3^-$ (approximately 1 μCi mL$^{-1}$ final concentration) and incubated in UVR

transparent Teflon-FEP bottles under full solar radiation exposures in a refrigerated tank contiguous to

the experimental microcosms. The teflon bottles were tied on top of a UVR transparent acrylic tray,

keeping all bottles under flat and constant position. Tray was wrapped with 2 layers of neutral density

screen to obtain saturating but non-photoinhibitory solar exposures. For analysis of the fraction of the

fixed carbon incorporated into particulate (POC) and dissolved (DOC) organic carbon, 5 mL samples

were filtered through 0.2 μm PC filters (25 mm diameter) under low pressure (50 mm Hg) after the light

incubation period, using 2 manifolds simultaneously (10 positions per manifold). POC was retained on

the filter while the filtrate was directly collected in scintillation vials to assess $^{14}C$ activity in the dissolved

fraction (DOC). Simultaneously, the total amount of organic carbon incorporated in the cells (TOC) was

measured independently of the DOC-POC filtration by processing 5 mL of the incubated samples. Non-

assimilated $^{14}C$ was released by exposing the filters (POC) to acid fumes (50% HCl) or by adding 200 μl

of 10% HCl to the liquid samples (DOC & TOC, respectively) and shaking overnight. The radioactivity

of each sample was measured using a Wallac WinSpectral 1414 scintillation counter (EG&G Company,

Finland). There was no significant difference between measurements of TOC compared to the sum of the

particulate and dissolved fractions ($R^2$=0.94, n=12).


*2.5 Bacterial production and respiration*

Bacterial production (BP) was measured following the [$^3$H] leucine incorporation method (Smith & Azam

1992). Three replicates and 1 killed control were sampled (1 mL) from each experimental unit on days 0,

1, 2, 3, 4 and 7 of the second incubation. Samples were spiked with 40 μL Leucine (47 μCi mL$^{-1}$ specific

activity stock solution), and incubated for 80 min in the same chamber and growth conditions than the



bacterioplankton assemblages. Processed samples were analyzed in a Wallac Win-Spectral 1414 scintillation counter and the BP was calculated from the Leucine uptake rates employing the theoretical leucine to carbon conversion factor (3.1 kg C mol Leu$^{-1}$) (Simon & Azam 1989).

Samples for bacterial respiration (BR) were taken on days 0, 2, 4 and 7 of the second incubation. BR in

each sample was determined from the difference of the dissolved oxygen concentration consumed between the end and the start of a 24 h dark incubation using 50 mL Winkler bottles in duplicate. The 24 h incubation was carried out at the same temperature than the experiment. Dissolved oxygen concentrations were determined by automated high-precision Winkler titration, using a potentiometric end point detector, Metrohm 721 DMS Titrino, as described in Serret et al. (1999). Bacterial carbon

demand (BCD) was calculated as the sum of BP and BR. Bacterial growth efficiency (BGE) was obtained from the proportion of the BCD that was used for bacterial production (BGE= BCD/BP).

*2.6 Flow cytometry analysis*

For phytoplankton (i.e. first incubation), samples were collected, fixed with P+G (1% paraformaldehyde

+ 0.05% glutaraldehyde) and analyzed with a FACScalibur flow cytometer (Becton–Dickinson). Measurements of the different photodetectors were made with a logarithmic amplification for each signal, and the trigger was set on red fluorescence (FL3). Phytoplankton counts were obtained at a high flow rate (1.05 µL s$^{-1}$) during 10 min. Two size groups of cells (R1 and R2) were discriminated on the bivariate plots of side light scatter (SSC) *vs.* FL3.

For bacterioplankton counts (i.e. second incubation), samples were stained with 2.5 µM of SYTO-13 (Molecular Probes) dissolved in dimethyl sulfoxide. The samples were incubated during 10 min at room

temperature in the dark, followed by the addition of 10 µL of a microspheres solution (FluoSpheres carboxylated-modified microspheres (1.0 µm nominal diameter). ThermoFisher Scientific) as internal standard for instrument performance. Samples were then immediately analyzed. The threshold was set on the green fluorescence (FL1). Stained bacteria were discriminated and counted in a bivariate plot of SSC *vs.* FL1.

Viability of bacterioplankton community was measured on day 7 using the 5-Cyano-2,3,-ditolyl tetrazolium chloride (CTC) dye (Sieracki et al. 1999; Gasol and Arístegui 2007). The CTC can freely diffuse into the cells where it is reduced by healthy respiring bacteria, producing a precipitated colored red/orange formazan product. This product is detectable and quantifiable by flow cytometry (Rodriguez et al. 1992). Samples were stained with 45 mM of CTC during 60 minutes and then analyzed. Threshold was set on the FL3 and viable bacteria were counted in a bivariate plot of SSC *vs.* FL3.

All data were acquired and analyzed with the software CellQuest (Becton– Dickinson) as Flow Cytometry Standard files.

*2.7 DOC concentration*

Dissolved organic carbon samples were taken in 250 mL acid-washed all-glass flasks and were gently filtered through acid rinsed 0.2 µm Pall-Supor filters. All this process was done in an acid-cleaned all-glass device under low $N_2$ flow pressure. About 10 mL of the filtrates were distributed in pre-combusted (450 ˚C for 24 h) glass ampoules acidified with 50 µL of 25% $H_3PO_4$. The ampoules were heat sealed immediately and stored at 4 ˚C until analysis with a Shimadzu TOC-VCS analyzer following the high temperature catalytic oxidation method (Álvarez-Salgado & Miller 1998).

*2.8 Statistical analysis*

When the data followed a normal distribution and homoscedasticity, tested by Lillieforst test, a one way-

ANOVA and multiple comparison post-test (MCP-test) or *t*-Test were employed to determine differences

among the mean of several or paired samples, respectively. In the case that data did not follow a normal

distribution, a non-parametric repeated-measures ANOVA (RMANOVA) and a Tukey-Kramer multiple

comparison post-test were chosen. The Wilcoxon signed rank test was used to compare non-parametric

paired samples. The confidence level was established at the 95%. Statistical analysis was performed using

the software packages MatLab R2012b and GraphPad InStat TM v2.04a+.

**3 Results:**

During the first incubation, aimed to obtain the organic matter inocula under current and future $CO_2$

conditions, the LC treatment $pCO_2$ values were close to the atmospheric equilibrium, with values ranging

between $419 \pm 21$ ppm $CO_2$ on day 0 and $226 \pm 38$ ppm $CO_2$ on day 3 (mean and SD, n=3) (Fig. 3A). In

the HC treatment $pCO_2$ values increased since the beginning of the incubation until reaching values

around 1200 ppm the last four days. Maximum values were observed at day 5 with $1227 \pm 149$ ppm $CO_2$.

Chl *a* used as an indicator of the phytoplankton biomass showed similar trends in the two treatments. It

increased with the initial addition of inorganic nutrients showing an early bloom on day 1 with Chl *a*

values of $21 \pm 4$ µg $L^{-1}$ and $22 \pm 9$ µg $L^{-1}$ for LC and HC treatments, respectively. Chl *a* concentration

decreased after the bloom, keeping values close to the lowest concentrations on day 6 for the HC treatment

with $3.0 \pm 0.4$ µg $L^{-1}$ and on day 5 for the LC treatment, $3.1 \pm 0.5$ µg $L^{-1}$ (Fig. 3A).

Flow cytometry results showed a succession of two different phytoplankton populations along the incubation characterized by differences in cell size and Chl content (Table 1). In both treatments, larger cells with higher Chl *a* fluorescence and cell complexity (Region 1, Table 4.2.1), dominated the phytoplankton community at the beginning of the incubation. The abundance of these large cells decreased, especially in the HC treatment, and at the end of the incubation smaller cells, with lower Chl content became more abundant (Region 2, Table 4.2.1). At day 7 the small size fraction population was dominant and the analyzed phytoplankton community was similar in both, HC and LC treatments.

Primary production rates followed the Chl *a* pattern with a marked peak blooming the first incubation day followed by a decline afterwards in all the treatments (Fig. 4). The increase in total carbon fixation during the bloom was due to an increase in both, DOC and POC production, but it was mainly due to POC assimilation. The percentage of extracellular release of dissolved carbon (PER=DOC/(POC+DOC)) ranged between 18% and 77%. DOC, POC, and therefore TOC, were higher during the bloom under the LC treatment but not significantly different than the rates observed in the HC treatment. Differences in the production rates between both treatments became negligible after the second incubation day (Fig. 4). DOC concentration increased from day 0 to maximum values on day 7, and similar to Chl *a* concentration, production rates and cell abundance, there were not significant differences between the two $CO_2$ treatments at the end of the incubation (Fig. 3B). Parallel analysis of DOM fluorescence (*i.e.* protein-like and humic-like substances) also supported the later results (data not shown).

In the second incubation, aimed to assess the effects of $CO_2$ and the organic matter additions on bacterioplankton, $pCO_2$ and pH were similar within the same $CO_2$ treatment (*i.e.* LC_NO and LC_AO *vs.* HC_NO and HC_AO), but significantly different between LC and HC treatments (Fig. 5A and B). $pCO_2$

in the LC ranged between $350 \pm 28$ ppm and $568 \pm 187$ ppm $CO_2$ (mean and SD, n=3) on day 5 and 2,

respectively. In contrast, HC treatments increased from $397 \pm 18$ ppm on day 0 to a maximum value of

$2213 \pm 229$ ppm on day 3, significantly different that the expected 1000 ppm $pCO_2$ in the HC treatments.

The maximum was followed by a pronounced decrease on day 4, and subsequently, the values were

similar to the bubbled air concentrations ($1011 \pm 75$ ppm on day 5) (Fig. 5A). As expected from the $pCO_2$,

pH values in the LC treatments were fairly constant with a mean value of 8.07 but decreased markedly

from 8.03 on day 0 to 7.51 on day 3 in the HC treatment. After this minimum, pH increased to values

around 7.8 until the end of the experiment (Fig. 5B).

The changes in $CO_2$ concentration and pH were concomitant with an increase in BR in the HC treatments

(Fig. 5C). BR was also fairly constant in the LC treatments, but increased significantly from day 0 to day

2 in the HC treatments. After reaching the maximum, bacterial respiration dropped to similar values than

those observed in the LC treatments (Fig. 5C). Among them the LC_NO treatment showed the lowest

variability in the respiration rates and the highest values at the end of the incubation. Similar to $pCO_2$ and

pH, statistical differences between samples with inocula produced under current and future $CO_2$ scenarios

within each $CO_2$ treatment were not significant regarding respiration rates. Flow cytometry analysis of

the bacteria viability using the CTC dye, which has been previously related to the respiration rates,

showed that the LC_NO treatment had significantly higher values than the other treatments on day 7

(RMANOVA and Tukey-Kramer MCT, $p<0.05$). In contrast, the other three treatments did not show

significant differences among them (Fig. 6)

The initial concentration of dissolved organic carbon (DOC, µM-C) in the samples was quite similar

among the four treatments until the peak in respiration. DOC before the addition of the organic matter

inocula was 89 µM-C and increased in all the treatments to maximum values of $111 \pm 5$ µM-C and $117 \pm$ n.d. µM-C in treatments LC_AO and HC_NO on day 3, respectively. Afterwards DOC in the LC treatments kept approximately constant but decreased 27% in the HC treatments compared to the LC treatments ($81 \pm 3$ µM-C and $81 \pm 2$ µM-C in HC_NO and HC_AO, respectively). Statistical analysis

showed significant differences between both $pCO_2$ treatments on day 7 (ANOVA and $t$-Test for paired treatments, p<0.01) (Fig. 7).

Unlike $pCO_2$, pH, respiration and DOC, bacterial abundance and production showed differences regarding the origin of the organic matter. The addition of the organic matter inocula produced a fast increase in production and abundance from day 0 to day 3 in all the treatments. Cell abundance increased from $6.2 \pm$

$0.4 \times 10^4$ bacteria mL$^{-1}$ before the addition of the organic matter inocula to a maximum of $9.0 \pm 0.4 \times 10^5$ bacteria mL$^1$ in the HC_AO treatment (Fig. 8A). Differences for the bacterial abundance started to be significant in day 1 but they were more clearly observed during the maximum at day 3 (ANOVA and MCP-test, p<0.05). Bacterial abundances were 29% and 31% higher in samples where the acidified organic matter was added than in those with the addition of organic matter produced under the current

CO$_2$ scenario, in the LC and HC treatments, respectively (Fig. 8A).

Additionally, bacterial production increased from a minimum value of $1.1 \pm 0.1$ µg C L$^{-1}$ d$^{-1}$ on day 0 to maximum values on day 2 for the four treatments, ranging between $185 \pm 37$ µg C L$^{-1}$ d$^{-1}$ and $208 \pm 4$ µg C L$^{-1}$ d$^{-1}$ for HC_NO and LC_NO, respectively. From the maximum values at day 2 to the end of the incubation, the treatments with the addition of acidified organic matter (LC_AO and HC_AO) showed a

higher decrease in the production rates than those with the addition of the non-acidified organic matter,

resulting in significant differences later on. On day 7 treatments LC_NO and HC_NO produced 53% and

45% more than treatments LC_AO and HC_AO, respectively (Fig. 8B).

The BCD was biased by the respiratory activity at the beginning of the incubation and by the production

at the end, showing the treatments with the addition on non-acidified organic matter significantly higher

carbon demand than the treatments with the addition of acidified organic matter at day 7 (ANOVA and

MCP-test, $p<0.05$) (Fig 9A). In consequence, the BGE was higher for the LC treatments, at the beginning

of the incubation during the activity peak, but decreased at the end. At day 7 the HC treatment with the

addition of non-acidified organic matter (HC_NO) showed significantly higher efficiency than the other

treatments (ANOVA and MCP-test, $p<0.05$) (Fig 9B). Conversely, the HC_AO treatment expected for

future scenarios of global change did not show significant differences with neither of the LC treatments.

## 4 Discussion:

The main goal of the current study was to distinguish between the direct and indirect effects of ocean

acidification on natural bacterial assemblages. To achieve this objective we performed a 2×2 experimental

design combining the acidification of seawater and the addition of phytoplankton-derived organic matter

produced under current and future $CO_2$ conditions and natural solar exposures. Although there have been

described different ways to modify the seawater pH to simulate an ocean acidification scenario, the

continuous bubbling of the natural plankton communities with a target $CO_2$ concentration of 1000 ppm

$CO_2$ was chosen in the present study to simulate the $pCO_2$ and pH conditions expected for the end of the

century. This method ensures quite realistic responses in terms of acclimation rates (Rost et al. 2008). It

also allows that changes in the biological activity of the samples enable the modification of the $pCO_2$

values if, for example, the photosynthetic or respiratory rates become faster than the rates needed to achieve the $CO_2$ chemical equilibrium in seawater. Changes in $pCO_2$ values due to microbial activity are usually observed in natural waters during bloom events in surface waters or in areas with high amount of organic matter (Joint et al. 2011). $pCO_2$ also increases with depth due to the increase in heterotrophic activity compared to the autotrophic activity in surface (Pukate & Rim-Rukeh 2008, Dore et al. 2009) and can change due to upwelling events, for example in the same area where the samples were collected, reaching values close to those observed in the present work (Alvarez et al. 1999, Gago et al. 2003).

In our study, the $CO_2$ enrichment did not produce a significant effect on phytoplankton production or biomass, measured as [14]C incorporation or Chl $a$ concentration, respectively. Phytoplankton community composition changed from bigger to smaller phytoplankton cells, as has been observed in similar microcosm studies (Reul et al. 2014, Grear et al. 2017), but differences between present and future $CO_2$ treatments were not observed. DOCp production increased during the bloom evolved at the beginning of the incubation but bulk DOC concentration showed similar values between $CO_2$ treatments, as expected based on the lack of differences observed in the biological and metabolic parameters. Despite several studies indicate an increase in phytoplankton carbon production and biomass under future scenarios of $CO_2$ (Kroeker et al. 2013), in our study exposure of the cells to natural conditions including solar UVR might have counteracted the stimulatory effect of the high $CO_2$, since it increases the sensitivity of phytoplankton to photoinhibition (Sobrino et al. 2008, 2009, Gao et al. 2012).

The addition of the organic matter and the start of the aeration in the second incubation produced a burst in the metabolic activity of the bacteria. Bacterioplankton growing in the HC treatments showed higher rates of respiration the first two days. This response was opposite to that described for some bacterial

cultures (Teira et al. 2012) and seems to be related to an acclimation to the new pH values, somehow

similar to observed for phytoplankton (Sobrino et al. 2008). Consequently, $pCO_2$ in the HC treatments

increased more than expected due to the biological activity carried out by bacterioplankton. The increase

in respiration was also concomitant to an increase in bacterial abundance and production. However, the

results indicate that while changes in respiration were related to $pCO_2$ values, changes in bacterial

abundance and production were mainly related to the origin of the organic matter amendments. The

bacterial abundance was stimulated by the presence of organic matter from a phytoplankton community

grown under high $CO_2$ conditions compared to the addition of organic matter values grown under current

$CO_2$ conditions, at the beginning of the incubation (up to day 3). On the other hand, differences in bacterial

production were only significant at the end of the experiment (day 7).

Possible explanations for the preference of the organic matter produced under future $CO_2$ conditions by

bacteria are changes in the composition and quality of this organic matter, changes in the capability of

bacteria to use this organic matter via extracellular enzymatic activity or a combination of both. Several

mesocosm studies have shown that the rate of extracellular release of phytoplankton dissolved organic

matter production and the formation of transparent exopolymer particles (TEPs) is higher under high $CO_2$

conditions (Engel 2002, Liu et al. 2010, Endres et al. 2014). TEPs are mostly carbon rich, can be easily

consumed by heterotrophic bacteria and can aggregate and coagulate in larger gel-like structures that

provide nutrients and attachment surface to bacteria, finally acting as hotspots where bacteria can grow

and develop (Azam & Malfatti 2007). However, our results from the initial incubation aimed to obtain

phytoplankton-derived organic matter inocula for bacteria do not suggest differences on DOC

concentration under current and future $CO_2$ conditions (Fig. 3B), and neither composition, since

phytoplankton community and fluorescence DOM properties were similar in both treatments at the end

of the experiment (Table 1). Recent results from mesocosm experiments showing no significant

differences in DOM concentration and composition between current and future $CO_2$ levels also

corroborate these findings (Zark et al. 2015). Other than this, very little was found in the literature about

the direct impact of ocean acidification on the DOM pool, particularly on its molecular composition and

long-term reactivity.

Despite the significant effect on bacterial abundance during the activity peak, bacterial production only

showed significant differences among treatments later during the incubation, having bacteria inoculated

with the organic matter produced under high $CO_2$ conditions lower production rates than those with the

organic matter produced under low $CO_2$ conditions. These results support a higher bioavailability of the

organic matter produced under high $CO_2$ conditions to bacteria earlier during the incubation, increasing

bacterial abundance, and decreasing production later at the end of the experiment. The lower DOC values

in the HC treatments at the end of the experiment compared to the LC treatments partially supports this

contention in the HC_AO treatment. However, they mainly indicate a clear effect of the pH,

independently of the origin of the organic matter inocula. We hypothesized that low pH, directly or

indirectly, could be responsible for a faster and more efficient degradation of the most recalcitrant organic

carbon, also in the HC_NO, at the end of the experiment. On one hand, the lower DOC in the HC

treatments compared to the LC treatments at the end of the experiment could be an indirect consequence

of the higher respiration rates, which lowered the pH values (i.e. $p$CO$_2$ values above the expected 1000

ppm) in these treatments during the activity peak. Between the two incubation performed for this study,

the significant effect of the pH on the DOC content might have been only observed in the second

incubation because it reached almost double de $pCO_2$ than the first one (i.e. min pH= 7.48 for the

bacterioplankton incubation *vs.* min pH= 7.71 for the phytoplankton incubation). Following this

contention, if acidification increases the degradation rate and bioavailability of the organic matter, that

would explain why respiration in the control treatment (LC_NO) was only significantly higher at the end

of the experiment, where neither the organic carbon added nor the environment were modified by the

acidification with $CO_2$. Flow cytometry results confirmed the highest rates of respiration in the LC_NO

treatment since the CTC dye is reduced to a quantifiable fluorescent CTC-formazan inside cells when the

electron transport system is active (Sieracki et al. 1999, Gasol & Arístegui 2007).

On the other hand, previous studies have confirmed that many enzymes involved in the hydrolysis of

organic matter are sensitive to pH variations. It has been reported that important enzymes for bacterial

metabolism such as leucine aminopeptidase or the *α*- and *β*-glucosidase enhance their activity and

transformation rates with small decreases in pH, similar to the values measured in this study (Grossart et

al. 2006, Piontek et al. 2010, 2013). The effects of ocean acidification on enzyme activity is not constant

and could vary depending on enzyme type and geographical location (Yamada & Suzumura 2010).

Despite our results show that the increase in bacterial abundance is mainly related to the presence of

organic matter produced under future $CO_2$ conditions and not so much to the decrease in pH, a trend that

supports a synergistic effect of both, acidified organic matter and pH, can be observed. This trend shows

the highest effects under the most extreme HC_AO treatment and the lowest effects under the less

aggressive LC_NO treatment (i.e. for bacterial abundance the 3[rd] day of incubation and partially for

production and viability on day 7). It can be also possible that enhanced extracellular enzymes produced

by the microbial community under high carbon conditions during the first incubation aimed to obtain the

organic matter inocula came with the acidified organic matter since there is evidence that $\beta$-glucosidase, leucine aminopeptidase, and phosphatase enzymes are stable in cold waters for weeks (Steen & Arnosti 2011).

The uncoupling between pH and organic matter effects on bacterial respiration and production made that

the carbon demand was biased by the most significant effect along the incubation. Therefore the least perturbed treatment with cells grown under present $CO_2$ conditions and non acidified organic mater (i.e. LC_NO) showed the highest growth efficiency after the organic matter addition, during the activity peak, because of the lack of pH changes and lower respiration rates. However, after 7 days an intermediate treatment with high $CO_2$ levels but non-acidified organic matter addition (i.e. HC_NO) had the highest

growth efficiency due to the higher production rates and the relatively lower respiration rates compared to the treatment grown under current conditions but addition of non-acidified organic matter (LC_NO). Cells grown under future scenarios with high $CO_2$ levels and acidified organic matter additions did not perform differently than those grown under present $CO_2$ conditions, independently of the addition of acidified or non-acidified organic matter. BGE and bacterial abundance did agree at the end of the

experiment showing the HC_NO treatment the highest cell abundance, although without significant differences among treatments. Nonetheless, further studies are needed to disentangle the lack of agreement among bacterial production, abundance and growth efficiency, more specially during the peaks of activity since they seem to show the biggest uncoupling between the measured parameters (del Giorgio & Cole 1998). Lack of total agreement regarding the statistical analysis might be related to differences in

methodological sensitivity and variability. Revision of leucine (or thymidine) to carbon ratios under



different $CO_2$ levels, which might be an important source of variation between bacterial production and abundance, should be also approached to enhance our understanding in this topic.

The results from this investigation show that ocean acidification can significantly affect bacterioplankton metabolism directly by changes in the respiration rate and indirectly by changes on the organic matter
with concomitant effects on bacterial production and abundance. They demonstrate that future scenarios of global change, with higher acidification might not result in a higher turnover of organic matter by bacteria.

*Acknowledgements:* This study was possible thanks to the financial support from Xunta de Galicia
(7MMA013103PR project and ED431G/06 Galician Singular Research Center) and Spanish Ministry of Economy, Industry and Competitiveness (CTM2014-59345-R project and H. Sanleón-Bartolomé fellowship (FPI-IEO fellowship 2011/06)), and the technical assistance from Maria José Fernández Pazó, Vanesa Vieitiez and ECIMAT staff. The authors are especially grateful to Xosé A. Alvarez-Salgado and Marta Álvarez for their scientific support and suggestions on the manuscript and to María Pérez Lorenzo and Marta Ruiz Hernández for their help with the
respiration and bacterial production techniques, respectively. The authors of this study do not have conflict of interest to declare.



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

Oceanogr Ser 65:346



Table 1: Phytoplankton abundance (cells mL$^{-1}$) measured by flow cytometry from phase 1 incubation

aimed to obtain the organic matter inocula under acidified and non-acidified conditions.

| | Phytoplankton abundance (cells mL$^{-1}$) | | | |
|---|---|---|---|---|
| | Region 1 | | Region 2 | |
| | HC | LC | HC | LC |
| Day 2 | 10968 ± 4313 | 6207 ± 1524 | 1683 ± 1011 | 1857 ± 599 |
| Day 5 | 833 ± 771 | 431 ± 43 | 1780 ± 1303 | 1227 ± 562 |
| Day 7 | 994 ± 151 | 1308 ± 500 | 4002 ± 1913 | 4790 ± 333 |
**6 Figure legends:**

Figure 1. Geographical location of Ría de Vigo in the NW Iberian Peninsula. The insert shows a more

detailed map of the Ría de Vigo and the locations of A) ECIMAT and B) sampling station.

Figure 2. Experimental design of the study.

Figure 3.  A) Phytoplankton biomass measured as Chl *a* concentration and $p$CO$_2$ evolution along the first

incubation period aimed to obtain the organic matter inocula under current and future CO$_2$ conditions.

Black and striped vertical bars correspond to the Chl *a* mean ± SD (n=3) (mg L$^{-1}$) obtained under high

and low carbon (HC and LC) treatments, respectively. Black and grey circles correspond to the pCO$_2$

mean ± SD (n=3) (ppm) in HC and LC treatments, respectively. B) Temporal evolution of the dissolved

organic carbon (DOC) concentration (µM – C) during this first incubation. The black and grey dots

indicate the mean ± SD (n=3) of DOC from HC and LC treatments, respectively

Figure 4. Primary production measured as the incorporation of $^{14}$C into organic compounds during the

first incubation period aimed to obtain the organic matter inocula under current and future CO$_2$ conditions

A) Total organic carbon obtained from a sample independently of POC and DOC sample processing

(TOC) B) Particulate organic carbon (POC) C) Dissolved organic carbon from phytoplankton origin

(DOCp). Mean ± SD (n=3).

Figure 5. A) $pCO_2$ (ppm) and B) pH values in the four treatments of the second incubation period, respectively. Mean ± SD (n=3). C) Temporal evolution of bacterial respiration ($\mu$mol $O_2$ $d^{-1}$) in the four treatments during the second incubation period, mean ± SD (n=3). In all figures, black and grey triangles represent HC_NO and HC_AO treatments, respectively. Black and grey circles correspond to LC_NO and LC_AO treatments.

Figure 6. Bacterial viability (Viable bacterial $mL^{-1}$) measured with the CTC dye on day 7, mean ± SD (n=3). The asterisk indicate mean significant difference with the other treatments with p-value < 0.05.

Figure 7. Temporal evolution of the dissolved organic carbon (DOC) concentration ($\mu$M – C) during the second incubation period. Black and grey triangles represent HC_NO and HC_AO treatments, respectively. Black and grey circles correspond to LC_NO and LC_AO treatments, respectively. The two asterisks indicate significant differences between LC vs. HC treatments with p-value < 0.01.

Figure 8. A) Histogram of bacterial abundance evolution (Cells $mL^{-1}$) in the four treatments during the course of the second incubation period, mean ± SD (n=3). Treatments with different letter indicate significant differences (p-value < 0.05). B) Bacterial production ($\mu$g C $L^{-1}$ $d^{-1}$) from the four treatments during the second incubation period, mean ± SD (n=6). Asterisks indicate significant differences between NO *vs.* AO treatments with p-value < 0.05. Black and grey triangles represent HC_NO and HC_AO treatments, respectively. Black and grey circles correspond to LC_NO and LC_AO treatments, respectively.



680

Figure 9. A) Bacterial carbon demand (BCD) and B) Bacterial growth efficiency (BGE) calculated from the bacterial production and respiration rates obtained from this study. Asterisks indicate significant differences with p-value $< 0.05$ between NO *vs.* AO treatments, for the BCD and between HC_NO *vs.* the rest of the treatments (LC_NO, LC_AO and HC_AO) for the BGE. Black and grey triangles represent HC_NO and HC_AO treatments, respectively. Black and grey circles correspond to LC_NO and LC_AO treatments, respectively.





FIGURE 1:

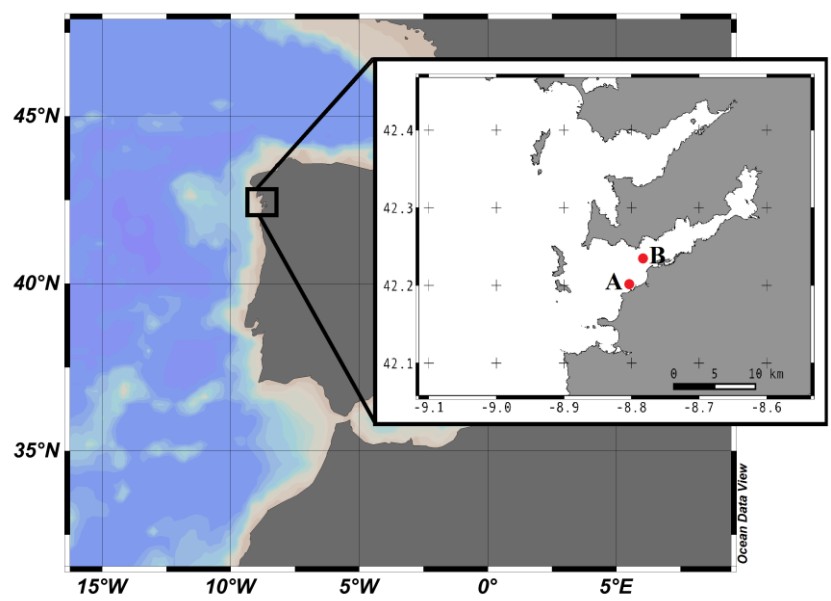



FIGURE 2:

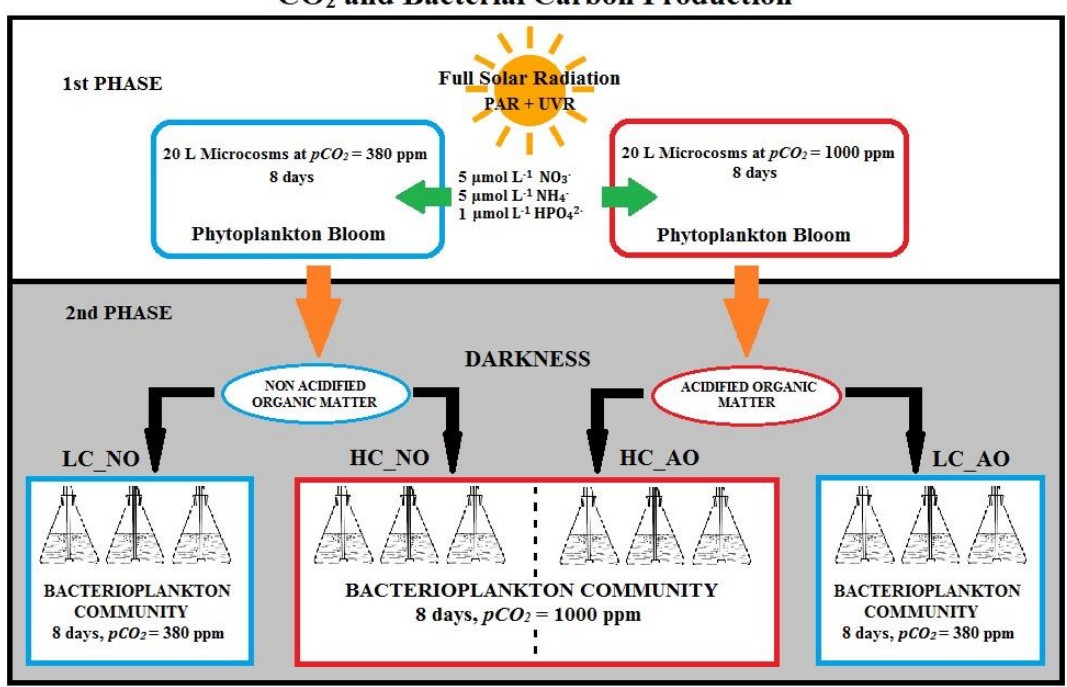





FIGURE 3:

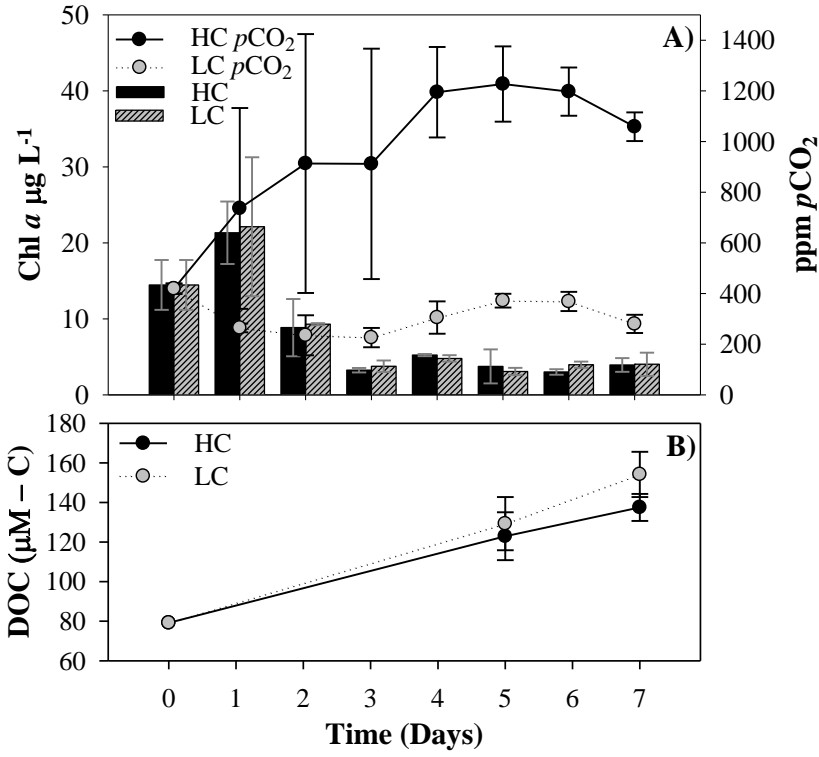



690    FIGURE 4.

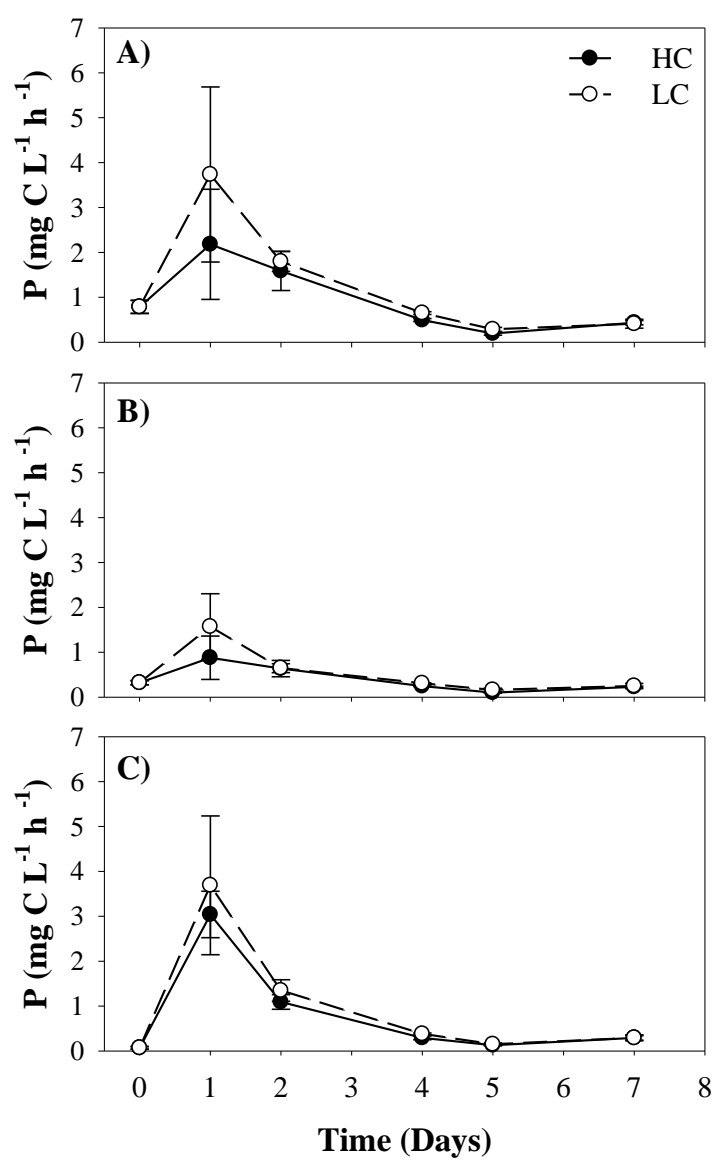



FIGURE 5:

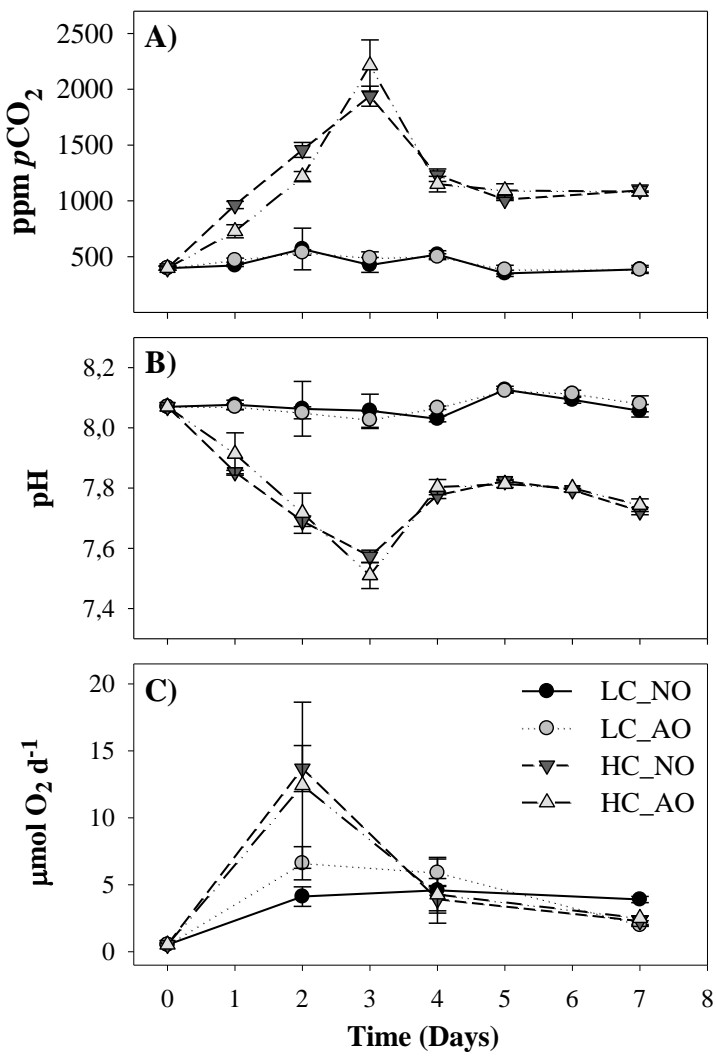





FIGURE 6:

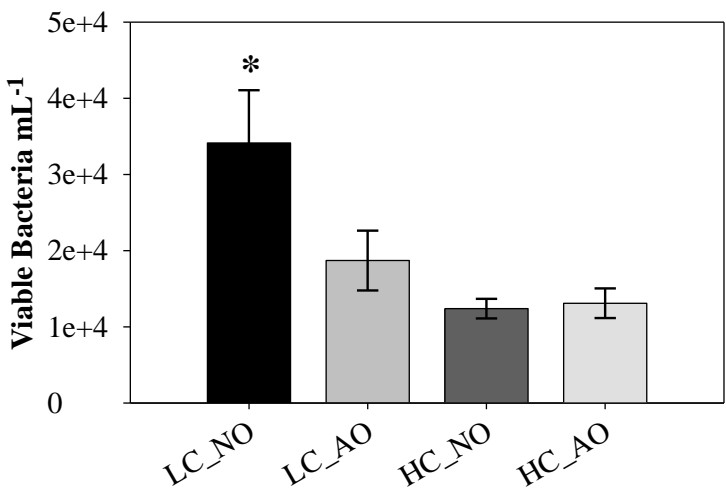



FIGURE 7:

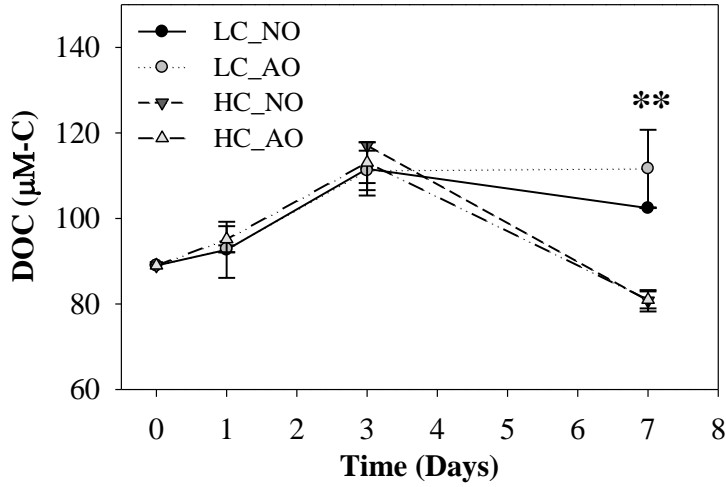



FIGURE 8:

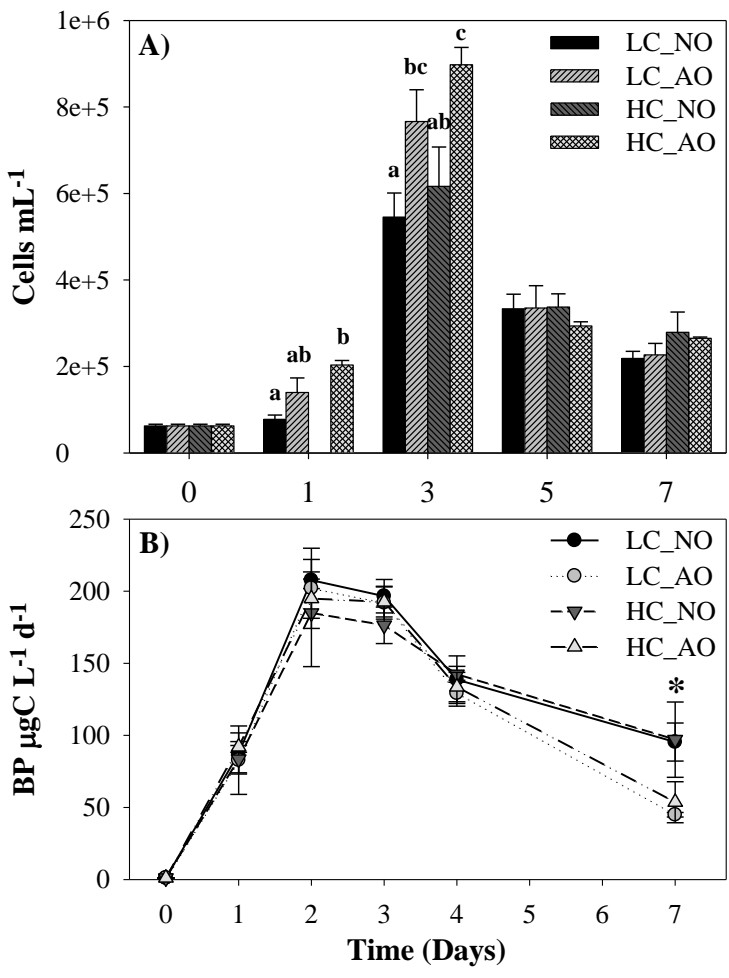



695     FIGURE 9:

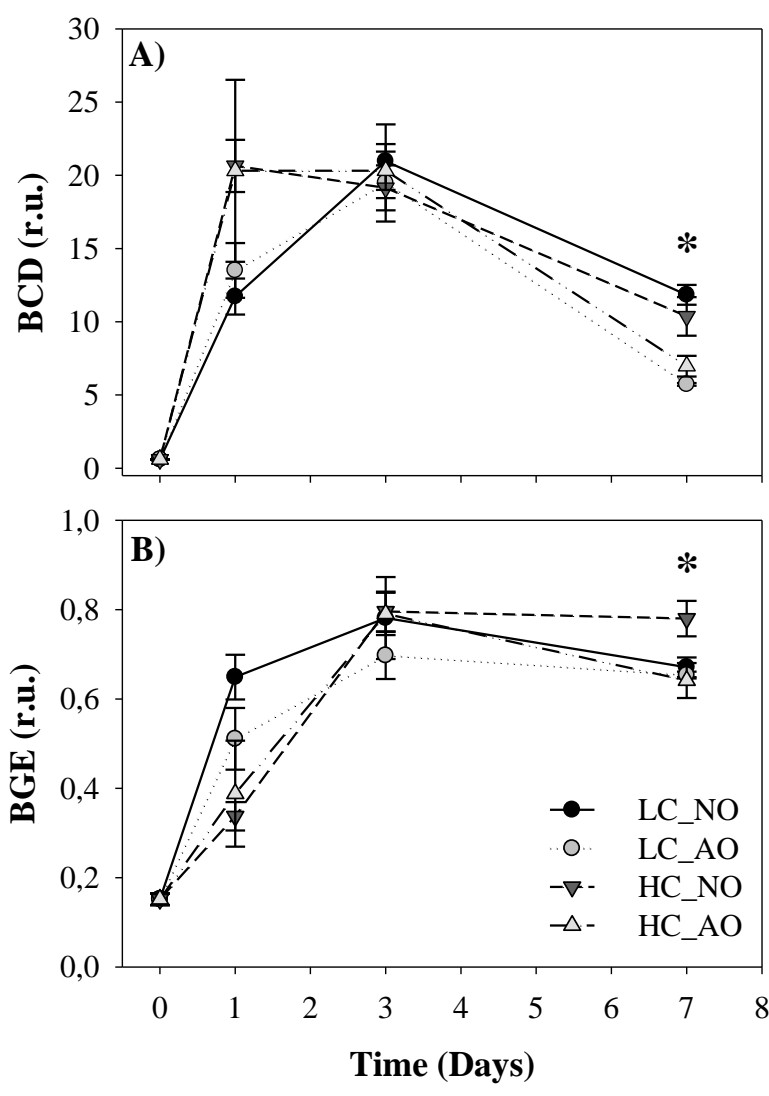