# Peer review of "INTERACTION BETWEEN ELEVATED CO₂ AND PHYTOPLANKTON-DERIVED ORGANIC MATTER UNDER SOLAR RADIATION ON BACTERIAL METABOLISM FROM COASTAL WATERS"

_Biogeosciences, 2017_

## Referee Comment (RC1) · Anonymous Referee #1 · 16 Feb 2018

Review of Fuentes-Lema et al titled Interaction between elevated CO2 and phytoplankton derived organic matter under solar radiation on bacterial metabolism from coastal waters

The paper presents data from an experiment performed in two stages. First, natural water was bubbled in present day and future CO2 conditions, and this water was using in the second stage of the experiment were a mix of this water, 1.2 $\mu$m filtered water with bacteria and 0.2 $\mu$m filtered water (without bacteria).

The first stage of the experiment produced non-acidified organic matter (NO) and acidified organic matter (AO). The second phase used the NO and AO treatments and exposed them to the same $CO_2$ conditions used in the first phase termed low carbon (LC) and high carbon (HC). All combinations were used, i.e. there were four treatments LC_NO, LC_AC, HC_NO and HC_AO, each with three replicates.

It took me some time to figure out what actually had been done in the experimental set-up, but Fig 2 is very good in outlining this. It is an interesting set up with the two stages which were used to see what the potential carry over effects of water grown under different $CO_2$ concentration may have on the bacterial production.

One possible bias is that the water produced in the first phase of the experiment was frozen until the start of the second phase of the experiment. Freezing the water might affect the DOM pool. This was the same for all treatments so I do not see this as a major flaw, but you could consider taking this aspect up in the discussion.

For the statistical test, why not use a two way ANOVA, you have two factors NO/AO and LC/HC? Also for the BR, BP I would suggest to do a regression of the development, e.g. on the cumulative respiration/production, then the whole time series could be taken into account. This way you could compare the cumulative results to the single point results.

I would also like to see a deeper analysis of what happens with the physiological variables over time. You found a higher respiration initially in the HC treatment, but this shifted towards highest respiration in the LC_NO at the end. Why is that and what are the different processes that could be involved? Similar with BP, it increases initially in all treatments, but in the end there is a clear difference between treatments. I know you take up some aspects e.g. the effect of pH on enzymatic activity, but there could be other aspects e.g. intracellular pH regulation, and the literature points to different directions.

Overall the manuscript is well written and easy to follow. The figures are of sufficient

quality and I have no problem supporting the publication after taking my suggestions above into consideration.

Minor comments:

Please add the actual p value throughout the results chapter where statistical tests were conducted, also where there is no statistical difference, i.e. $p < 0.05$ is not acceptable; the limit should be at $p < 0.001$ or $p < 0.0001$, so less than 0.1% or 0.01% probability for a type II error.

In line 475 you write: On one hand (should be: on the one hand), and it is only in line 490 as a start of a new paragraph where you have the follow up: on the other hand . . . Please rephrase, these two points should come right after each other if you want to keep them in the 'one the one hand', 'on the other hand' form

y-axis in Figs 5 and 9 has ',' as decimal separator

Please add the axis title, for example Respiration to Fig 5c with units in parenthesis. It makes it much easier to see what data is presented.

---

## Referee Comment (RC2) · Anonymous Referee #2 · 3 Apr 2018

General comments The paper by Fuentes-Lema and co-workers addresses a topic of interest in marine biogeochemistry. The experimental design is appropriate although the results are far from concluding and I feel the authors have overexploited a bit their findings. Some of the conclusions do not hold or do so only for one of the sampling points, which diminishes the overall relevance of their contribution. I think they should be much more cautious in some statements. Another problem that complicates the interpretation of the dataset is that not all variables were sampled at the same time (e.g. bacterial abundance data are lacking on days 2, 4 and 6 and respiration is lacking on

days 1, 3, 5 and 6). This makes the assessment of the effect of high CO2 concentrations on DOM-heterotrophic bacteria interactions very difficult. I suggest an alternative approach. Rather than focusing on the analysis of specific sampling times I would like to see the analysis of integrated values of bacterial biomass, production and respiration over the course of the 7 days of the incubation of the second phase. Maybe the conclusions will change but they will be more reliable than in the current version. In general, the paper is well-written although there are a number of instances in which English usage and grammar needs to be improved.

Specific comments An important concern is related to the authors' point about the lability of DOC. By examining their Figure 7 one cannot really say anything about DOC lability since only in the HC treatments there was a net, although very slight, decrease in DOC concentration, presumably due to bacterial uptake. How can the authors explain the general pattern of increase rather than decrease in DOC for most of the experiment?

The manuscript implicitly assumes that UV played a distinct role in the amount and quality of DOC produced by phytoplankton (DOCp) but there is no way of distinguishing the effect of UV from the effect of PAR in their experimental design.

By incubating the samples in the dark the authors are introducing a potential source of error in their results. I fully concur with them that solar radiation plays an important role in DOM-microbial plankton interactions but then, why stop the normal diel cycle of light and darkness during 8 days? The authors should be aware of the possible role of photoheterotrophy in bacterioplankton communities (e.g. Ruiz-González et al. 2013 Frontiers in Microbiology) and their response to the two types of DOM. Moroever, the DOM enriched seawater could also be subject to further transformations caused purely by sunlight that are not accounted for in their setup.

Since no attempt was made to estimate empirical leucine-to-carbon conversion factors for calculating bacterial production, known to change dramatically in different environmental conditions (see for instance Teira et al. 2015 Applied and Environmental Microbiology, and references therein), presumably met during their second phase incubations, the uncertainties of this variable (BP) and that of bacterial carbon demand (BCD) are very high.

Information about phytoplankton cell counts of two idly defined groups (Region 1 and Region 2 in Table 1 for which we do not even know their sizes) in flow cytometry analysis, assuming that the huge initial (Day 0-1) increase in chlorophyll a concentration was mostly due to large cells not detected by flow cytometry make this section virtually irrelevant. Also, I guess that Synechococcus cyanobacteria were surely present at least in Day 2, with abundances much higher than 1000-10000 cells mL-1. The authors should ellaborate more on these results or simply delete them.

Technical The title is very confusing. The interaction is established between DOM and bacteria, not between elevated CO2 and phytoplankton-derived DOM. Also, what does it mean "Interaction. . . on bacterial metabolism"? The expression "Under solar radiation" is not necessary to be included in the title. "bacterial metabolism from coastal waters" also reads awkwardly. Please change to a more informative, correct title.

L. 54-55. "Phytoplankton" and "heterotrophic counterparts" are not logical choices. Please refer to autotrophic and heterotrophic microbes or something similar.

L. 60. What do the authors refer by "The other way round"? Please explain.

L. 64-65. Provide more detail about "the abiotic AND biotic factors".

L. 67. "Adaptation towards a fast acclimation" sounds odd. The underlying mechanisms are different, please rephrase.

L. 103. "subjected. . . concentrations" can be safely eliminated here.

L. 142. Surely there were other phytoplanktonic taxa/groups present along with "maily diatoms".

L. 194. The R2 value does not inform about the significance of this difference. Did the aothors performed a t-test/one-way ANOVA to support their statement?

L. 206. Duplicate Winkler bottles seem too few for oxygen changes measurements. Usually a minimum of 4-5 bottles are used.

L. 249. "to compare non-parametric paired samples" is an odd phrasing.

L. 293. It does not seem so obvious to me.

L. 302. Why using RMANOVA for comparing differences at one single sampling point?

L. 310. I do not follow the rationale for using the two statistical tests here. There is some confusion about statistics throughout the manuscript. The authors should clearly state which tests they used and why or try a different analysis (see my general comment) with changes integrated over the course of the incubation of phase 2.

L. 328. "biased" is probably not the best word here.

L. 329. Replace "on" by "of".

L. 341. "there have been...simulate" reads awkward. Please rephrase.

L. 345. Are the authors sure of this statement?

L. 370-371. This is not true in view of the different sampling points and the data shown in the corresponding figures.

L. 396-397. "having... production rates" is not correct English usage.

L. 398-400. Pleaase see my previous comment about lability.

L. 432. Do the authors imply that their water samples collected on June 27th were "cold"? Maybe there was a strong upwelling on that day but this information is not provided and ca. 15°C is not exactly cold.

L. 454. Respiration rate and organic matter are not independent.

L. 455. These results are far from "demonstrating" that claim.

I am not conviced that "acidified organic matter" and "non acidified organic matter" are the best terms for their treatments, did they check that the resultant DOM was of lower pH in the former treatment? Fig. 5B simply shows that the sample water had a lower pH but not that the DOM was indeed of lower pH.

$\mu$M is not the appropriate SI unit, please correct it to $\mu$mol L-1.

Fig. 4. Please replace the "P" in the Y-axes by TOC, POC and DOC. This is not a very relevant figure and can be eliminated or moved to supplementary information.

The authors use total abundance of bacteria but probably data about the contribution of low and high nucleic acid content (LNA and HNA) cells are available, as well as some indication of changes in cell size that would provide a good estimate of bacterial biomass that could be compared with changes in BP, even if they were assuming data from the literature to convert from leucine incorporation rates to carbon units.

Dubbing cells able to reduce CTC as "viable" is not the best term. Most authors, including the cited references, refer to them as cells actively respiring or showing active respiration but the number of viable cells in their incubations was likely much higher, just by comparing the cell abundance numbers of Figs. 6 and 8. It is uncommon to show CTC positive cells before total bacterial abundance. Also, why not showing the dynamics of CTC positive bacteria for the entire experiment rather than only at day 7?

Fig. 9B. BGE is given either as a percentage or as a ratio, what does "r.u." mean? Also, given the use of a very high and constant leucine-to-carbon conversion factor of 3.1 kg C mol Leu-1, if the values are close to 80% BGE, they are extremely high, very difficult to reconcile with almost no net uptake of DOC (Fig. 7).

---

## Author Comment (AC1) · 6 Jun 2018

**We are very grateful to anonymous referee #1 for the positive evaluation of our work and the constructive suggestions and comments, which have certainly contributed to improve the manuscript. We have followed all her/his indications and have made the suggested modifications in the manuscript.**

General comments:

- 1. The paper presents data from an experiment performed in two stages. First, natural water was bubbled in present day and future $CO_2$ conditions, and this water was using in the second stage of the experiment were a mix of this water, 1.2 μm filtered water with bacteria and 0.2 μm filtered water (without bacteria). The first stage of the experiment produced non-acidified organic matter (NO) and acidified organic matter (AO). The second phase used the NO and AO treatments and exposed them to the same $CO_2$ conditions used in the first phase termed low carbon (LC) and high carbon (HC). All combinations were used, i.e. there were four treatments LC_NO, LC_AC, HC_NO and HC_AO, each with three replicates.

It took me some time to figure out what actually had been done in the experimental set-up, but Fig 2 is very good in outlining this. It is an interesting set up with the two stages which were used to see what the potential carry over effects of water grown under different $CO_2$ concentration may have on the bacterial production.

One possible bias is that the water produced in the first phase of the experiment was frozen until the start of the second phase of the experiment. Freezing the water might affect the DOM pool. This was the same for all treatments so I do not see this as a major flaw, but you could consider taking this aspect up in the discussion.

Some information about the possible effects and rationale of freezing the samples between phase 1 and 2 has been included in the text to address this methodological issue with more detail, as proposed by the referee. (Lines 363-372)

We agree with the referee that some alterations of the dissolved organic matter pool can occur during the freezing process but we think they should not be significant for the objectives of this study (see explanation below). In addition, this methodological approach was the best for the objectives of our study due to the following reasons: 1) Freezing the samples was necessary to preserve the acidified and non acidified organic matter for later use and to avoid the excessive consumption of the organic matter between phase 1 and 2 in our experiment. DOC released by marine phytoplankton is highly labile to bacterial utilization and can be degraded significantly within hours (Chen & Wangersky 1996). 2) Freezing the samples instead keeping them refrigerated or filtered decreased significantly the survival rate of the microbial community from phase 1 (Postgate & Hunter 1961) and allowed to keep most of the organic matter, including both, the particulate and dissolved organic matter fractions. We thought that keeping both, the particulate and dissolved organic matter fractions was a more realistic experimental approach since the particulate fraction could potentially be degraded during the second phase of the experiment due to the elevated $CO_2$ concentrations (Piontek et al. 2010).

Regarding DOC preservation methods, unfortunately there is not too much available information for marine samples. Some of them include the acidification with clorhydric or phosphoric acid, or the poisoning of the samples with mercuric chloride solution, which are obviously not appropriate for our experimental design (Chen & Wangersky 1996, Griffith & Raymond 2011, Calleja et al. 2013). However, results from peatland waters show that changes in DOC concentration by freeze/thaw cycles are small (<5%) and do not show a clear pattern of increase or decrease (Peacock et al. 2015). In any case, such changes are in the range of typical precision for DOC analysis and, as indicated by the referee, all samples were processed similarly in our experiment, so we think that the freezing and thawing process was not a major problem for achieving the objectives of the study.

- 2. For the statistical test, why not use a two way ANOVA, you have two factors NO/AO and LC/HC? Also for the BR, BP I would suggest to do a regression of the development, e.g. on the cumulative respiration/production, then the whole time series could be taken into account. This way you could compare the cumulative results to the single point results.

**A new approach using a two way ANOVA statistical test has been done and has been included in the present version of the manuscript, as proposed by the referee. A detailed explanation of this topic has been included in the Statistical analysis section (Lines 222-229).**

**Regarding the second suggestion proposed by the referee about using regressions of the cumulative values, both referees (Referee#1 and Referee#2) showed the importance of using a cumulative approach to improve the understanding of the results. We also agree that this approach improve the quality of the manuscript helping to understand the results. In this sense, we decided to use the integral of the cumulative values, as proposed by referee#2, instead the regression because it calculates the exact value under the cumulative response curve. In the new version of the manuscript we have included some text showing the results of the cumulative responses and statistical analysis (Lines 279, 299, 304 and 315) for the bacterial production, bacterial respiration, bacterial abundance, bacterial carbon demand and the bacterial growth efficiency and the corresponding figures (Fig. 4D, 7B, 7D, 8B and 8D). The results from this methodological approach do not change the conclusions shown in the previous version of the manuscript but they do certainly help to make the reading and understanding of the results much easier.**

- 3. I would also like to see a deeper analysis of what happens with the physiological variables over time. You found a higher respiration initially in the HC treatment, but this shifted towards highest respiration in the LC_NO at the end. Why is that and what are the different processes that could be involved? Similar with BP, it increases initially in all treatments, but in the end there is a clear difference between treatments. I know you take up some aspects e.g. the effect of pH on enzymatic activity, but there could be other aspects e.g. intracellular pH regulation, and the literature points to different directions.

**Additional information regarding other possible mechanisms recently shown by the literature (i.e. Bunse et al. (2016)) has been included as proposed by the referee (Line 415).**

Overall the manuscript is well written and easy to follow. The figures are of sufficient quality and I have no problem supporting the publication after taking my suggestions above into consideration.

Minor comments:

4. Please add the actual p value throughout the results chapter where statistical tests were conducted, also where there is no statistical difference, i.e. $p < 0.05$ is not acceptable; the limit should be at $p < 0.001$ or $p < 0.0001$, so less than 0.1% or 0.01% probability for a type II error.

**All the exact p-values, together with the n value and the statistical test used, have been included in the new version of the manuscript as proposed by the referee.**

- 5. In line 475 you write: On one hand (should be: on the one hand), and it is only in line 490 as a start of a new paragraph where you have the follow up: on the other hand…. Please rephrase, these two points should come right after each other if you want to keep them in the 'one the one hand', 'on the other hand' form.

**The text has been corrected as proposed.**

- 6. y-axis in Figs 5 and 9 has ',' as decimal separator.

**All figures have been revised and corrected.**

- 7. Please add the axis title, for example Respiration to Fig 5c with units in parenthesis. It makes it much easier to see what data is presented.

**Axis titles have been included for improving clarity as indicated by the referee.**

**A NEW VERSION OF THE MANUSCRIPT, INCLUDING ALL THE SUGGESTIONS RAISED BY THE REVIEWER, IS INCLUDED AT THE END OF THE REFEREE´S RESPONSE AS A POSSIBLE NEW VERSION OF THE MANUSCRIPT.**

**References:**

Bunse C, Lundin D, Karlsson CMG, Vila-Costa M, Palovaara J, Akram N, Svensson L, Holmfeldt K, González JM, Calvo E, Pelejero C, Marrasé C, Dopson M, Gasol JM, Pinhassi J (2016) Response of marine bacterioplankton pH homeostasis gene expression to elevated $CO_2$. Nat Clim Chang 6:483

Calleja ML, Batista F, Peacock M, Kudela R, McCarthy MD (2013) Changes in compound specific $\delta 15N$

amino acid signatures and d/l ratios in marine dissolved organic matter induced by heterotrophic bacterial reworking. Mar Chem 149:32–44

Chen W, Wangersky PJ (1996) Rates of microbial degradation of dissolved organic carbon from phytoplankton cultures. J Plankton Res 18:1521–1533

Griffith DR, Raymond PA (2011) Multiple-source heterotrophy fueled by aged organic carbon in an urbanized estuary. Mar Chem 124:14–22

Peacock M, Freeman C, Gauci V, Lebron I, Evans CD (2015) Investigations of freezing and cold storage for the analysis of peatland dissolved organic carbon (DOC) and absorbance properties. Environ Sci Process Impacts 17:1290–1301

Piontek J, Lunau M, Handel N, Borchard C, Wurst M, Engel A (2010) Acidification increases microbial polysaccharide degradation in the ocean. Biogeosciences 7:1615–1624

Postgate JR, Hunter JR (1961) On the Survival of Frozen Bacteria. J Gen Microbiol 26:367–378

**EFFECTS OF ELEVATED $CO_2$ AND PHYTOPLANKTON-DERIVED ORGANIC MATTER ON THE METABOLISM OF BACTERIAL COMMUNITIES FROM COASTAL WATERS**

Antonio Fuentes-Lema[1]; Henar Sanleón-Bartolomé[2]; Luis M. Lubián[3][†]; Cristina Sobrino[1][*]

[1] - UVigo Marine Research Centre; Lagoas Marcosende Campus 36331 Vigo, Spain

[2] - Spanish Institute of Oceanography (IEO). Paseo Marítimo Alcalde Francisco Vázquez 10, 15001 A Coruña, Spain.

[3] - Institute of Marine Sciences of Andalucía (CSIC). Campus Univ. Rio San Pedro. 11519 Puerto Real, Cádiz, Spain

*Correspondence to*: Cristina Sobrino. +34 986 818789, sobrinoc@uvigo.es

Running head: Ocean acidification and DOM on bacteria

**Abstract.** Microcosm experiments to assess bacterioplankton response to phytoplankton-derived organic matter obtained under current and future-ocean $CO_2$ levels were performed. Surface seawater enriched with inorganic nutrients was bubbled for 8 days with air (current $CO_2$ scenario) or with a 1000 ppm $CO_2$–air mixture (future $CO_2$ scenario) under solar radiation. The organic matter produced under the current and future $CO_2$ scenarios was subsequently used as inoculum. Triplicate 12 L flasks filled

with 1.2 µm-filtered natural seawater enriched with the organic matter inocula were incubated in the dark for 8 days under $CO_2$ conditions simulating current and future $CO_2$ scenarios to study the bacterial response. The acidification of the media increased bacterial respiration at the beginning of the experiment while the addition of the organic matter produced under future levels of $CO_2$ was related to changes in bacterial production and abundance. This resulted in 67% increase in the integrated bacterial respiration under future $CO_2$ conditions compared to present $CO_2$ conditions and 41% higher integrated bacterial abundance with the addition of the acidified organic matter compared to samples with the addition of non acidified organic matter. This study demonstrates that the increase in atmospheric $CO_2$ levels can affect bacterioplankton directly by changes in the respiration rate and indirectly by changes on the organic matter with concomitant effects on bacterial production and abundance.

KEY WORDS: bacterioplankton, phytoplankton, organic matter, ocean acidification.

**1 Introduction**:

The increase in fossil fuel burning, cement production and deforestation together with changes in land use have resulted in an accumulation of atmospheric $CO_2$ at levels never seen before the last two million years (Caldeira & Wickettt 2008, Le Quere et al. 2015). Atmospheric gases can freely diffuse into the ocean surface, which has already absorbed about 30% of the emitted anthropogenic $CO_2$, perturbing the carbonate system and decreasing ocean pH in a process known as ocean acidification (Sabine et al. 2004, Burke et al. 2014).

The latest IPCC report shows that the pH of surface ocean waters has already decreased by 0.1, corresponding to a 26% increase in acidity. If mitigation strategies for global change are not adopted and $CO_2$ emissions continue as usual, ocean pH values will drop about 0.3-0.7 units by the end of the 21$^{st}$ century (Burke et al. 2014). The decrease in seawater pH has strong effects on the ecosystem, the aquatic organisms and the interactions among them. Studies about its consequences in the surface ocean have been primarily focused on calcifying organisms such as corals or coralline algae because they participate in the formation of habitats and human services (Langdon et al. 2003, Fabry et al. 2008). Recent meta-analysis studies also revealed decreased survival, growth, development and abundance of a broad range of marine organisms, although the magnitude of these responses varies between taxonomic groups, including variation within similar species (Kroeker et al. 2013). Additionally, other authors have demonstrated that ocean acidification can increase growth, primary production and $N_2$ fixation rates in some phytoplankton species (Barcelos e Ramos et al. 2007, Fu et al. 2007, Levitan et al. 2007, Iglesias-Rodriguez et al. 2008). In contrast with the abundant information about phytoplankton, very little is known about the response of bacterioplankton.

Heterotrophic bacteria play an important role in the planktonic community since they are responsible for the majority of the organic matter remineralization (Cole et al. 1988, Azam et al. 1998, Nagata et al.

65    2000) allowing the primary producers to make use of the recycled inorganic nutrients. They also return dissolved organic carbon (DOC) to the marine food web via its incorporation into bacterial biomass through what it is called the microbial loop (Azam et al. 1983). However, the microbial response can change depending on phytoplankton taxonomic composition and the nutrient levels, and therefore productivity, of the water (Teira et al. 2012, Bunse et al. 2016, Sala et al. 2016). Despite this important

70    role of bacterioplankton in the marine food web and biogeochemical cycles, only few studies have been designed to elucidate the effects of ocean acidification on bacteria metabolism or its interaction with the abiotic (i.e. temperature, ultraviolet radiation, mixing, etc.) and biotic factors (i.e. algal derived organic matter, microbial community composition, trophic interactions, etc.). Some of these studies suggest an absence of significant metabolic responses in experiments where $CO_2$ levels were manipulated

75    (Rochelle-Newall et al. 2004, Allgaier et al. 2008, Newbold et al. 2012). Fast acclimation of the metabolic machinery to low pH values might have occurred in those experiments 
[revised manuscript text omitted]
 the organic matter produced under current $CO_2$ conditions in the previous incubation (from herein named as Non acidified Organic matter, NO, n=6) and the other half with the organic matter produced under future $CO_2$ conditions by acidifying the media as explained above (from herein named as Acidified Organic matter, AO, n=6). In each case, three replicates were aerated with ambient $CO_2$ levels (Low Carbon treatment (LC): LC_NO, LC_AO) or air with 1000 ppm $CO_2$ (High Carbon treatment (HC): HC_NO, HC_AO). This experimental setup produced 4 different treatments from the less modified sample (LC_NO) to the most altered sample (HC_AO) (Fig. 2). The bottles were located in a walk-in growth chamber under dark conditions at 15 ºC, similar to the *in situ* temperature. The dark incubation for the bacterioplankton samples was aimed to focus the experiment on the two factors of the study, $CO_2$ and organic matter addition, avoiding potential effects of solar radiation on bacterioplankton and organic matter that might complicate the interpretation of the results.

*2.2 DIC, pH and $CO_2$ analysis*

[revised manuscript text omitted]

*2.7 Statistical analysis*

Data followed a normal distribution and homoscedasticity, tested by Jarque-Bera test and Bartlett test, respectively. Thus, a parametric one-way ANOVA or two-way ANOVA statistical tests were used to determine differences between one or two experimental factors, respectively (i.e. $CO_2$ in the first incubation, $CO_2$ and organic matter addition in the second incubation). If the interaction between the two factors was significant a multiple comparison post-test (MCP-test) was carried out. The confidence level was established at the 95%. Statistical analysis was performed using the software package MatLab R2012b.

**3 Results:**

During the first incubation, aimed to obtain the organic matter inocula under current and future $CO_2$ conditions, the LC treatment $pCO_2$ values were close to the atmospheric equilibrium, with values ranging between $419 \pm 21$ ppm $CO_2$ on day 0 and $226 \pm 38$ ppm $CO_2$ on day 3 (mean and SD, n=3) (Fig. 3A). In the HC treatment $pCO_2$ values increased since the beginning of the incubation until reaching values around 1200 ppm the last four days. Maximum values were observed at day 5 with $1227 \pm 149$ ppm $CO_2$. Chl *a* used as an indicator of the phytoplankton biomass showed similar trends in the two treatments. It increased with the initial addition of inorganic nutrients showing an early bloom

on day 1 with Chl *a* values of 21 ± 4 µg L$^{-1}$ and 22 ± 9 µg L$^{-1}$ for LC and HC treatments, respectively.

320 Chl *a* concentration decreased after the bloom, keeping values close to the lowest concentrations on day 6 for the HC treatment with 3.0 ± 0.4 µg L$^{-1}$ and on day 5 for the LC treatment, 3.1 ± 0.5 µg L$^{-1}$ (Fig. 3A). Analysis of phytoplankton composition by flow cytometry did not show significant differences in community structure between LC and HC treatments (data not shown).

Primary production rates followed the Chl *a* pattern with a marked peak blooming the first incubation

325 day followed by a decline afterwards in all the treatments (Figure 1 in Appendix 1). The increase in total carbon fixation during the bloom was due to an increase in both, POC and DOCp production, but it was mainly due to POC assimilation. The percentage of extracellular release of dissolved carbon (PER=DOCp/(POC+DOCp)) ranged between 18% and 77%. DOCp, POC, and therefore TOC, were higher during the bloom under the LC treatment but not significantly different than the rates observed in

330 the HC treatment (1-way ANOVA, TOC: p=0.307, POC: p=0.242, DOCp: p=0.527, n=6). Differences in the production rates between both treatments became negligible after the second incubation day (Figure 1 in Appendix 1).

Bulk DOC concentration increased from day 0 to maximum values on day 7, and similar to Chl *a* concentration and production rates, there were not significant differences between the two $CO_2$

335 treatments at the end of the incubation (1-way ANOVA, p=0.096, n=6) (Fig. 3B). Parallel analysis of DOM fluorescence (*i.e.* protein-like and humic-like substances) also supported the later results (data not shown).

In the second incubation, aimed to assess the effects of $CO_2$ and the organic matter additions on bacterioplankton, $pCO_2$ and pH were similar within the same $CO_2$ treatment (*i.e.* LC_NO and LC_AO

*vs.* HC_NO and HC_AO), but significantly different between LC and HC treatments (Fig. 4A and B).

355 $p$CO$_2$ in the LC ranged between $350 \pm 28$ ppm and $568 \pm 187$ ppm CO$_2$ (mean and SD, n=3) on day 5 and 2, respectively. In contrast, HC treatments increased from $397 \pm 18$ ppm on day 0 to a maximum value of $2213 \pm 229$ ppm on day 3, significantly different that the expected 1000 ppm $p$CO$_2$ in the HC treatments. The maximum was followed by a pronounced decrease on day 4, and subsequently, the values were similar to the bubbled air concentrations ($1011 \pm 75$ ppm on day 5) (Fig. 4A). As expected

360 from the $p$CO$_2$, pH values in the LC treatments were fairly constant with a mean value of 8.07 but decreased markedly from 8.03 on day 0 to 7.51 on day 3 in the HC treatment. After this minimum, pH increased to values around 7.8 until the end of the experiment (Fig. 4B).

The changes in CO$_2$ concentration and pH were accompanied with an increase in BR in the HC treatments (Fig. 4C). BR was also fairly constant in the LC treatments, but showed a pronounced

365 increase from day 0 to day 2 being significantly different in the HC treatments (2-way ANOVA, p=0.048, n=12). After reaching the maximum, bacterial respiration dropped to similar values than those observed in the LC treatments (Fig. 4C). Among the LC treatments, the LC_NO treatment showed the lowest variability in the respiration rates and the highest values at the end of the incubation (1-way ANOVA and MCP-test, p<0.001, n=6). Flow cytometry analysis of the CTC-positive bacteria, which

370 have been previously related to actively respiring bacteria, also showed that the LC_NO treatment had significantly higher values than the other treatments on day 7 (1-way ANOVA and MCP-test, p=0.007, n=6). In contrast, the other three treatments did not show significant differences among them (Fig. 5). Similar to $p$CO$_2$ and pH, statistical differences between samples with organic matter inocula produced under current and future CO$_2$ scenarios within each CO$_2$ treatment were not significant

380 regarding respiration rates. The analysis of the cumulative responses, measured as the integrated bacterial respiration during the whole incubation, showed 67% higher values under HC than under LC conditions (2-way ANOVA, p=0.010, n=6) (Fig. 4D).

The initial concentration of dissolved organic carbon (DOC, $\mu$mol C L$^{-1}$) in the samples was quite similar among the four treatments until the peak in respiration (Fig. 6). DOC before the addition of the

385 organic matter inocula was 89 $\mu$mol C L$^{-1}$ and increased in all the treatments to maximum values of 111 $\pm$ 5 $\mu$mol C L$^{-1}$and 117 $\pm$ n.d. $\mu$mol C L$^{-1}$ in treatments LC_AO and HC_NO on day 3, respectively. Afterwards DOC in the LC treatments kept approximately constant but decreased 27% in the HC treatments compared to the LC treatments (81 $\pm$ 3 $\mu$mol C L$^{-1}$ and 81 $\pm$ 2 $\mu$mol C L$^{-1}$ in HC_NO and HC_AO, respectively). Statistical analysis showed significant differences between both $p$CO$_2$

390 treatments on day 7 (2-way ANOVA, p<0.001, n=12) (Fig. 6).

Unlike $p$CO$_2$, pH, respiration and DOC, bacterial abundance and production showed differences regarding the origin of the organic matter. The addition of the organic matter inocula produced a fast increase in production and abundance from day 0 to day 3 in all the treatments. Cell abundance increased from 6.2 $\pm$ 0.4 x 10$^4$ bacteria mL$^{-1}$ before the addition of the organic matter inocula to a

395 maximum of 9.0 $\pm$ 0.4 x 10$^5$ bacteria mL$^1$ in the HC_AO treatment (Fig. 7A). Differences for the bacterial abundance started to be significant in day 1 (1-way ANOVA and MCP-test, p=0.012, n=6) but they were more clearly observed during the maximum at day 3 (1-way ANOVA and MCP-test, p<0.001, n=6). Bacterial abundances were 29% and 31% higher in samples where the acidified organic matter was added than in those with the addition of organic matter produced under the current CO$_2$

400 scenario, in the LC and HC treatments, respectively (Fig. 7A). This resulted in 41% higher integrated

bacterial abundance with the addition of the acidified organic matter than in samples with the addition of non acidified organic matter (2-way ANOVA, p<0.001, n=12) (Fig. 7B).

Additionally, bacterial production increased from a minimum value of $1.1 \pm 0.1$ µg C L$^{-1}$ d$^{-1}$ on day 0 to maximum values on day 2 for the four treatments, ranging between $185 \pm 37$ µg C L$^{-1}$ d$^{-1}$ and $208 \pm 4$ µg C L$^{-1}$ d$^{-1}$ for HC_NO and LC_NO, respectively. However, the analysis of the integrated response did not show significant differences among treatments (Fig 7.D). Only punctually, the treatments with the addition of acidified organic matter (LC_AO and HC_AO) showed a higher decrease in the production rates than those with the addition of the non-acidified organic matter, resulting in significant differences later on (2-way ANOVA, p=0.001, n=12). On day 7 treatments LC_NO and HC_NO produced 53% and 45% more than treatments LC_AO and HC_AO, respectively (Fig. 7C).

The BCD was affected by the respiratory activity at the beginning of the incubation (day 1, 2-way ANOVA, p=0.041, n=12) and by the production at the end (day 7, 2-way ANOVA, p=0.002, n=12) (Fig 8A). In consequence, the BGE was higher for the LC treatments, at the beginning of the incubation during the activity peak, but decreased at the end. At day 7 the HC treatment with the addition of non-acidified organic matter (HC_NO) showed significantly higher efficiency than the other treatments (1-way ANOVA and MCP-test, p=0.002, n=6) (Fig 8C). Overall, integrated BCD resulted 24% higher under HC conditions while the BGE was 11% lower in the same conditions, compared to LC conditions (2-way ANOVA, p=0.011, p=0.019, respectively, n=6) (Fig 8B, D).

**4 Discussion:**

The main goal of the current study was to distinguish between the direct and indirect effects of ocean acidification on natural bacterial assemblages. To achieve this objective we performed a 2×2 experimental design combining the acidification of seawater and the addition of phytoplankton-derived organic matter produced under current and future $CO_2$ conditions and natural solar exposures. UVR induces photomineralization of coloured dissolved organic matter (CDOM) increasing the biological availability of the resulting DOM (Moran & Zepp 1997) and can also increase DOCp production in surface waters (Carrillo et al. 2002, Helbling et al. 2013, Fuentes-Lema et al. 2015).

Although there have been described different methodologies to modify the seawater pH to simulate an ocean acidification scenario, the continuous bubbling of the natural plankton communities with a target $CO_2$ concentration of 1000 ppm $CO_2$ was chosen in the present study to simulate the $pCO_2$ and pH conditions expected for the end of the century. This method simulates the natural variations of sea surface $pCO_2$ driven by differences between atmospheric and sea water $CO_2$ concentration providing quite realistic responses of the organisms to this environmental factor (Rost et al. 2008). It also allows that changes in the biological activity of the samples enable the modification of the $pCO_2$ values if, for example, the photosynthetic or respiratory rates become faster than the rates needed to achieve the $CO_2$ chemical equilibrium in seawater. Changes in $pCO_2$ values due to microbial activity are usually observed in natural waters during bloom events in surface waters or in areas with high amount of organic matter (Joint et al. 2011). $pCO_2$ also increases with depth due to the increase in heterotrophic activity compared to the autotrophic activity in surface (Pukate & Rim-Rukeh 2008, Dore et al. 2009) and can change due to upwelling events, for example in the same area where the samples were

collected, reaching values close to those observed in the present work (Alvarez et al. 1999, Gago et al. 2003).

In our study, the $CO_2$ enrichment did not produce a significant effect on phytoplankton production or biomass, measured as $^{14}C$ incorporation or Chl *a* concentration, respectively. Phytoplankton community composition changed from bigger to smaller phytoplankton cells, as has been observed in similar microcosm studies (Reul et al. 2014, Grear et al. 2017), but differences between present and future $CO_2$ treatments were not observed. DOCp production increased during the bloom evolved at the beginning of the incubation but bulk DOC concentration showed similar values between $CO_2$ treatments, as expected based on the lack of differences observed in the biological and metabolic variables. Despite several studies indicate an increase in phytoplankton carbon production and biomass under future scenarios of $CO_2$ (Kroeker et al. 2013), in our study exposure of the cells to natural conditions including solar UVR might have counteracted the stimulatory effect of the high $CO_2$, since it increases the sensitivity of phytoplankton to photoinhibition (Sobrino et al. 2008, 2009, Gao et al. 2012).

The addition of the organic matter and the start of the aeration in the second incubation produced a burst in the metabolic activity of the bacteria. However this increase in bacterial activity was not followed by a significant decrease in DOC. Taking into account the experimental design of the study keeping both, the particulate and the dissolved fractions of the organic matter from the first phase incubation, it is likely that production of DOC exceeded its consumption by heterotrophic bacteria, leading to a net accumulation of DOC in the microcosms. The particulate fraction could potentially have been degraded during the second phase of the experiment due to the elevated $CO_2$ concentrations or the enzymatic activity (Piontek et al. 2010). Enzymatic activity on POC depends on the type of substrate and relates to

the metabolic capabilities of the microbial community but can act on the temporal interval of our experiment (Arnosti et al. 2005). Counterintuitively, the high metabolic activity observed after adding the organic matter might have also contributed to the net accumulation of DOC since it has been recently demonstrated that microheterotrophs can release part of the substrate as new DOM (Lønborg et al. 2009). Freezing the sample with the organic matter produced in the first incubation might have also contributed to the DOC production during the second incubation. Freezing was necessary to avoid an excessive consumption of the acidified and non acidified organic matter for later use, since DOC released by marine phytoplankton is highly labile to bacterial utilization and can be degraded significantly within hours (Chen & Wangersky 1996). However, results from Peacock et al. (2015) suggest that changes in DOC concentrations by freeze/thaw cycles in samples stored no longer than six months are less than 5% and do not show a clear pattern of increase or decrease (Peacock et al. 2015). Such changes are within the range of typical precision for DOC analysis and should not be significantly different among the experimental treatments since all the samples were processed in a similar way.

Bacterioplankton growing in the HC treatments showed higher rates of respiration the first two days. This response was opposite to that described for some bacterial cultures (Teira et al. 2012) and seems to be related to an acclimation to the new pH values, somehow similar to observed for phytoplankton (Sobrino et al. 2008). Consequently, $p$CO$_2$ in the HC treatments increased more than expected due to the biological activity carried out by bacterioplankton. The increase in respiration was also parallel to an increase in bacterial abundance and production. However, the results indicate that while changes in respiration were related to $p$CO$_2$ values, changes in bacterial abundance were mainly related to the origin of the organic matter amendments. The bacterial abundance was stimulated by the presence of

[revised manuscript text omitted]

615 environmental conditions (Teira et al. 2015) and bacteria cultures under controlled conditions in the lab have shown that $CO_2$ fixation can increase under high carbon conditions, representing 8 to 9% of the bacterial production (Teira et al. 2012). All these factors can affect the bacterial production calculation. Despite our work used a conversion factor that had being empirically tested for natural samples collected from the same area without significant differences between stations with different $CO_2$

620 concentrations (Martínez-García et al. 2010), other factors such as the bacterial composition or the nature of organic matter additions might be affecting the final production rates. An attempt to empirically estimate the leucine-to-carbon conversion factors should be addressed in order to decrease uncertainties related to the bacterial production and the BCD and finally enhance our understanding in this topic.

625 In summary, the results from this investigation show that ocean acidification can significantly affect bacterioplankton metabolism directly by changes in the respiration rate and indirectly on bacterial abundance by changes in the organic matter.

>

*Acknowledgements:* This study was possible thanks to the financial support from Xunta de Galicia

630 (7MMA013103PR project and ED431G/06 Galician Singular Research Center) and Spanish Ministry of

Economy, Industry and Competitiveness (CTM2014-59345-R project and H. Sanleón-Bartolomé fellowship (FPI-IEO fellowship 2011/06)), and the technical assistance from Maria José Fernández Pazó, Vanesa Vieitiez and ECIMAT staff. The authors are especially grateful to Xosé A. Alvarez-Salgado and Marta Álvarez for their scientific support and suggestions on the manuscript and to María Pérez Lorenzo and Marta Ruiz Hernández for

640  their help with the respiration and bacterial production techniques, respectively. The authors of this study do not have conflict of interest to declare.

**6 Figure legends:**

 Figure 1. Geographical location of Ría de Vigo in the NW Iberian Peninsula. The insert shows a more detailed map of the Ría de Vigo and the locations of A) ECIMAT and B) sampling station.

Figure 2. Experimental design of the study.

 Figure 3. A) Phytoplankton biomass measured as Chl *a* concentration and $pCO_2$ evolution along the first incubation period aimed to obtain the organic matter inocula under current and future $CO_2$ conditions. Black and striped vertical bars correspond to the Chl *a* mean ± SD (n=3) ($\mu g\ L^{-1}$) obtained under high and low carbon (HC and LC) treatments, respectively. Black and grey circles correspond to the $pCO_2$ mean ± SD (n=3) (ppm $CO_2$) in HC and LC treatments, respectively. B) Temporal evolution

 of the dissolved organic carbon (DOC) concentration ($\mu mol\ C\ L^{-1}$)) during this first incubation. The black and grey dots indicate the mean ± SD (n=3) of DOC from HC and LC treatments, respectively.

Figure 4. A) $pCO_2$ (ppm $CO_2$) and B) pH values in the four treatments of the second incubation period, respectively. Mean ± SD (n=3). C) Temporal evolution of bacterial respiration ($\mu mol\ O_2\ d^{-1}$) and D)

 cumulative bacterial respiration ($\mu mol\ O_2$) during the second incubation period in the four treatments, mean ± SD (n=3). In all figures, black and grey triangles represent HC_NO and HC_AO treatments, respectively. Black and grey circles correspond to LC_NO and LC_AO treatments. The asterisk indicate

significant differences between LC *vs*. HC treatments with p-value < 0.05. The three asterisks indicate a significant difference between LC_NO with respect to the other treatments with p-value < 0.001.

860

Figure 5. Bacterial viability (Active bacterial mL$^{-1}$) measured with the CTC dye on day 7, mean ± SD (n=3). The two asterisks indicate mean significant difference with the other treatments with p-value < 0.01.

865 Figure 6. Temporal evolution of the dissolved organic carbon (DOC) concentration (µmol C L$^{-1}$) during the second incubation period, mean ± SD (n=3). Black and grey triangles represent HC_NO and HC_AO treatments, respectively. Black and grey circles correspond to LC_NO and LC_AO treatments, respectively. The three asterisks indicate significant differences between LC *vs*. HC treatments with p-value < 0.001.

870

Figure 7. A) Histogram of bacterial abundance evolution (Cells mL$^{-1}$), treatments with different letter indicate significant differences (p-value < 0.01), and B) cumulative response of bacterial abundance (Cells mL$^{-1}$) in the four treatments during the course of the second incubation period, mean ± SD (n=3).C) Bacterial production (µg C L$^{-1}$ d$^{-1}$) and D) cumulative bacterial production (µg C L$^{-1}$) from the 875 four treatments during the second incubation period, mean ± SD (n=3). The two asterisks indicate significant differences between NO *vs*. AO treatments with p-value < 0.01. Black and grey triangles represent HC_NO and HC_AO treatments, respectively. Black and grey circles correspond to LC_NO and LC_AO treatments, respectively.

880 Figure 8. Temporal evolution of A) bacterial carbon demand (BCD ($\mu$mol C L$^{-1}$ d$^{-1}$)), B) cumulative BCD ($\mu$mol C L$^{-1}$), C) bacterial growth efficiency (BGE) and D) cumulative BGE during the second incubation calculated from the bacterial production and respiration rates obtained from this study. One asterisk indicate significant differences with p-value < 0.05 between HC *vs.* LC treatments. Two asterisks indicate significant differences with p-value < 0.01 between NO *vs* AO treatments, for the

885 BCD and between HC_NO *vs.* the rest of the treatments (LC_NO, LC_AO and HC_AO) for the BGE. Black and grey triangles represent HC_NO and HC_AO treatments, respectively. Black and grey circles correspond to LC_NO and LC_AO treatments, respectively.

FIGURE 1:

[Figure]

FIGURE 2:

[Figure]

FIGURE 3:

[Figure]

[Figure]

FIGURE 5:

[Figure]

895     FIGURE 6:

[Figure]

[Figure]

FIGURE 8:

[Figure]

**7. Appendix 1:**

*Primary production*

Incubations were performed at noon and lasted 3 to 3.5 hours. Fifteen mL samples of each microcosm were inoculated with $H^{14}CO_3^-$ (approximately 1 µCi mL$^{-1}$ final concentration) and incubated in UVR transparent Teflon-FEP bottles under full solar radiation exposures in a refrigerated tank contiguous to the experimental microcosms. The teflon bottles were tied on top of a UVR transparent acrylic tray, keeping all bottles under flat and constant position. Tray was wrapped with 2 layers of neutral density screen to obtain saturating but non-photoinhibitory solar exposures. For analysis of the fraction of the fixed carbon incorporated into particulate (POC) and dissolved (DOCp) organic carbon, 5 mL samples were filtered through 0.2 µm PC filters (25 mm diameter) under low pressure (50 mm Hg) after the light incubation period, using 2 manifolds simultaneously (10 positions per manifold). POC was retained on the filter while the filtrate was directly collected in scintillation vials to assess $^{14}C$ activity in the dissolved fraction (DOCp). Simultaneously, the total amount of organic carbon incorporated in the cells (TOC) was measured independently of the DOCp-POC filtration by processing 5 mL of the incubated samples. Non-assimilated $^{14}C$ was released by exposing the filters (POC) to acid fumes (50% HCl) or by adding 200 µl of 10% HCl to the liquid samples (DOCp & TOC, respectively) and shaking overnight. The radioactivity of each sample was measured using a Wallac WinSpectral 1414 scintillation counter (EG&G Company, Finland). Data analysis determined that both, TOC and POC+DOCp results, were significantly correlated (y= 1.05x (±0.07) + 0.05 (±0.03), $R^2$=0.85, ANOVA, p=0.637, n=34).

920 Figure 1. Primary production rates measured as the incorporation of $^{14}C$ into organic compounds during the first incubation period aimed to obtain the organic matter inocula under current and future $CO_2$ conditions A) Total organic carbon fixation rates obtained from a different sample than the sample used for the determination of POC and DOCp fixation rates (TOC) B) Particulate organic carbon fixation rates (POC) C) Rates of dissolved organic carbon production from phytoplankton origin (DOCp). Mean

925 ± SD (n=3).

---

## Author Comment (AC2) · 6 Jun 2018

**Authors want to thank anonymous referee #2 for her/his deep review and helpful suggestions on the manuscript. We have followed all her/his indications and have made the suggested modifications in the manuscript. Each comment has been addressed point by point as shown in the text below.**

General comments:

- 1. The paper by Fuentes-Lema and co-workers addresses a topic of interest in marine biogeochemistry. The experimental design is appropriate although the results are far from concluding and I feel the authors have overexploited a bit their findings. Some of the conclusions do not hold or do so only for one of the sampling points, which diminishes the overall relevance of their contribution. I think they should be much more cautious in some statements. Another problem that complicates the interpretation of the dataset is that not all variables were sampled at the same time (e.g. bacterial abundance data are lacking on days 2, 4 and 6 and respiration is lacking on days 1, 3, 5 and 6). This makes the assessment of the effect of high $CO_2$ concentrations on DOM-heterotrophic bacteria interactions very difficult. I suggest an alternative approach. Rather than focusing on the analysis of specific sampling times I would like to see the analysis of integrated values of bacterial biomass, production and respiration over the course of the 7 days of the incubation of the second phase. Maybe the conclusions will change but they will be more reliable than in the current version. In general, the paper is well-written although there are a number of instances in which English usage and grammar needs to be improved.

**We agree with the referee that the analysis of the data can be improved if a more holistic approach is used in addition to analyzing the days where the responses showed statistical differences, which aimed to explain changes in the physiological variables over time. Following his/her suggestions, in the new version of the manuscript we have included new results from the statistical analysis of the integral values obtained from the cumulative responses (Lines 279, 299, 304 and 315) and the corresponding figures (Fig. 4D, 7B, 7D, 8B and 8D). The results from this methodological approach**

do not change the conclusions shown in the previous version of the manuscript but they do certainly help to make the understanding of the results much easier.

We have also carefully revised all the statements that might be controversial to avoid misunderstandings and the English usage and grammar has been corrected, as suggested by the referee.

Specific comments:

- 2. An important concern is related to the authors' point about the lability of DOC. By examining their Figure 7 one cannot really say anything about DOC lability since only in the HC treatments there was a net, although very slight, decrease in DOC concentration, presumably due to bacterial uptake. How can the authors explain the general pattern of increase rather than decrease in DOC for most of the experiment?

The experimental approach used for the study aimed to keep both fractions of the organic matter: the particulate and the dissolved organic matter fractions. We thought that keeping both, the particulate and dissolved organic matter fractions was a more realistic experimental approach since the particulate fraction could potentially be degraded during the second phase of the experiment due to the elevated $CO_2$ concentrations or the enzymatic activity (Piontek et al. 2010). Enzymatic activity on POC depends on the type of substrate and relates to the metabolic capabilities of the microbial community but can act on the temporal interval of our experiment (Arnosti et al. 2005). Therefore, it is likely that at the start of the bacterial incubation, production of DOC exceeded its consumption by heterotrophic bacteria, leading to a net accumulation of DOC in the microcosms. Counterintuitively, the high metabolic activity observed after adding the organic matter might have also contributed to the net accumulation of DOC since it has been recently demonstrated that microheterotrophs can release part of the substrate as new DOM, with a production efficiency of 11 ± 1%, 18 ± 2% and 17±2% for DOC, DON and DOP, respectively (Lønborg et al. 2009). Information regarding possible explanations for the increase in net DOC has been included in the text (Lines 353-363).

- 3. The manuscript implicitly assumes that UV played a distinct role in the amount and quality of DOC produced by phytoplankton (DOCp) but there is no way of distinguishing the effect of UV from the effect of PAR in their experimental design.

The fact that UVR plays an important role on DOC produced by phytoplankton (DOCp) has been already demonstrated in previous papers cited in the text (i.e. Carrillo et al. 2002, Helbling et al. 2013, Fuentes-Lema et al. 2015). Taking this into account we decided to expose the phytoplankton

**community during the first phase incubation to the whole solar spectra, including UVR. Since most of the experimental studies do not usually use UVR transparent materials, or do not take into account the effect of UVR at all, we though that it was important to recall this fact along the text, when it was relevant. However, to study the effect of UVR on phytoplankton DOC was not among the objectives of the study. In order to avoid confusion regarding the role of UVR as an additional experimental factor in our study, the reference to solar radiation shown in the title has been deleted. In addition the information related to the effect of UVR on DOM production from phytoplankton has been deleted from the Introduction. The new version of the manuscript only shows a brief comment regarding the importance of UVR on DOM production from phytoplankton later in the Discussion section (Lines 323-326).**

- 4. By incubating the samples in the dark the authors are introducing a potential source of error in their results. I fully concur with them that solar radiation plays an important role in DOM-microbial plankton interactions but then, why stop the normal diel cycle of light and darkness during 8 days? The authors should be aware of the possible role of photoheterotrophy in bacterioplankton communities (e.g. Ruiz-González et al. 2013 Frontiers in Microbiology) and their response to the two types of DOM. Moroever, the DOM enriched seawater could also be subject to further transformations caused purely by sunlight that are not accounted for in their setup.

**We agree with the referee that the effect of UVR on DOM-microbial plankton interactions is a relevant issue to be studied. However, we rather focused our attention on the other two factors (i.e. $CO_2$ and organic matter addition) that have been much less studied and therefore could bring more original information. Exposing the bacterioplankton samples to solar radiation, without testing the effect of UVR as an additional experimental factor, would have complicated significantly the interpretation of the results, and adding a third factor to the second phase incubation of our experimental design was logistically impossible. A third experimental factor with three replicates per treatment would have duplicated our experimental design from 12 to 24 experimental units, which was unavoidable by the authors. Just as an example, an experimental approach with 24 experimental units used in a microcosm study aimed to test the effect of $CO_2$ x Irradiance x nutrients addition on the microbial plankton responses was successfully carried out during the Group of Aquatic Productivity Workshop in Málaga in 2012 thanks to the collaboration of 23 researchers (ej. Sobrino et al. 2014, Mercado et al. 2014). The new version of the manuscript include a brief comment about the concern of using a third experimental factor (Lines 148-151).**

- 4. Since no attempt was made to estimate empirical leucine-to-carbon conversion factors for calculating bacterial production, known to change dramatically in different environmental conditions (see for instance Teira et al. 2015 Applied and Environmental Microbiology, and references therein), presumably met

during their second phase incubations, the uncertainties of this variable (BP) and that of bacterial carbon demand (BCD) are very high.

**The determination of the leucine-to-carbon conversion factors for calculating bacterial production under different $CO_2$ scenarios and organic matter additions is certainly an interesting study to be carried out. Unfortunately most of the researchers use a theoretical factor since it is sometimes unapproachable to determine them for each experimental condition, time and taxonomical composition, more especially when working with incubations of natural samples. In our experimental design we used a conversion factor, that despite being theoretical, it had been previously determined in an experiment aimed to study the differential responses of phytoplankton and heterotrophic bacteria to organic and inorganic nutrient additions in Ría de Vigo (Martínez-García et al. 2010). In this paper empirical Leu to C conversion factors for natural plankton communities from two different sampling sites of the Ría de Vigo, characterized by having different $CO_2$ concentrations (Ría and shelf) (Gago et al. 2003), was determined. The inner station (Ría) is the same sampling site than the location chosen for our study. Martínez-García et al. (2010) shows that no significant differences in conversion factors were found between both sampling locations (t-test, $p > 0.05$) or between the control and the treatment with organic matter additions. This lack of differences could imply an absence of significant effect of $CO_2$ concentration or organic matter addition on the conversion factor in our experiment. In addition, the conversion factor used in our study has been previously used in a paper aimed to analyze the effects of different $p$CO_2 concentrations on marine bacteria (Teira et al. 2012), which allows comparison with previously published results. However due to the lack of further empirical information supporting or rejecting the suitability of the conversion factors used in the manuscript, the authors have decided to delete most of the information related to the analysis of the bacterial carbon demand and bacterial growth efficiency, following the referee´s suggestion. The information has been deleted in the Abstract and in the Discussion sections, and it has been significantly decreased in the Results section. Information about the rationale for choosing the Leu to carbon conversion factor used in our study and some text about the uncertainties raised up by the referee related to the conversion factor, including new references from Martínez-García et al. (2010) and Teira et al. (2015), have been included in the text (Lines 179-185 and 460-469).**

- 5. Information about phytoplankton cell counts of two idly defined groups (Region 1 and Region 2 in Table 1 for which we do not even know their sizes) in flow cytometry analysis, assuming that the huge initial (Day 0-1) increase in chlorophyll a concentration was mostly due to large cells not detected by flow cytometry make this section virtually irrelevant. Also, I guess that *Synechococcus* cyanobacteria were surely present at least in Day 2, with abundances much higher than 1000-10000 cells mL-1. The authors should ellaborate more on these results or simply delete them.

**The purpose of including these flow cytometry results in the manuscript was to show that the phytoplankton composition in both CO$_2$ treatments during the first phase incubation, aimed to obtain the organic matter that was going to be used for the second phase incubation, was similar. These findings, in accordance with the lack of differences between treatments for primary production, are related to the absence of significant differences in DOM concentration at the end of the first incubation. In any case, since this information is more straightforward shown by the PP and DOM concentration, and the flow cytometry results do not represent the whole phytoplankton community, the table has been deleted as proposed by the referee.**

Technical:

- 6. The title is very confusing. The interaction is established between DOM and bacteria, not between elevated CO2 and phytoplankton-derived DOM. Also, what does it mean "Interaction: : : on bacterial metabolism"? The expression "Under solar radiation" is not necessary to be included in the title. "bacterial metabolism from coastal waters" also reads awkwardly. Please change to a more informative, correct title.

**A new more concise title has been written following the suggestions raised by the referee (Line 1).**

Minor comments:

- 7. L. 54-55. "Phytoplankton" and "heterotrophic counterparts" are not logical choices. Please refer to autotrophic and heterotrophic microbes or something similar.

**The text has been modified for clarification (line 53).**

- 8. L. 60. What do the authors refer by "The other way round"? Please explain.

**That was a misunderstanding. The sentence has been rephrased (line 59).**

- 9. L. 64-65. Provide more detail about "the abiotic AND biotic factors".

**The text has been modified for clarification (line 63).**

- 10. L. 67. "Adaptation towards a fast acclimation" sounds odd. The underlying mechanisms are different, please rephrase.

**The sentence has been modified to show the referee's suggestion (line 67).**

- 11. L. 103. "subjected… concentrations" can be safely eliminated here.

**The text indicated by the referee has been eliminated (line 97).**

- 12. L. 142. Surely there were other phytoplanktonic taxa/groups present along with "mainly diatoms".

**Due to the lack of data to corroborate this statement and that the sentence does not provide highly relevant information, the sentence has been deleted following the referee´s concern.**

- 13. L. 194. The R2 value does not inform about the significance of this difference. Did the authors performed a t-test/one-way ANOVA to support their statement?

**Information about the results from the one-way ANOVA statistical test has been included in Appendix 1 (Line 732). The analysis of the correlation between TOC and POC+DOC showed no significant differences (ANOVA, p=0.637, n=34) as indicated in the previous version of the manuscript.**

- 14. L. 206. Duplicate Winkler bottles seem too few for oxygen changes measurements. Usually a minimum of 4-5 bottles are used.

**Results for the analysis of bacterial respiration in our study come from the mean ± SD of 3 replicates (i.e. 12 L PC bottles) for each of the 4 treatments (LC_NO, LC_AO, HC_NO, HC_AO). For each replicate 4 Winkler bottles were used: Two (2) bottles to assess the $O_2$ concentration at the start of**

the incubation and another 2 to quantify the concentration at the end of the incubation. This makes a total of 12 bottles per treatment. Regarding variability within each technical replicate (i.e. duplicates from each 12 L PC bottles ), the error was very low, with an average of 1.7% (max: 8.6%, min: 0.06) and 1.3% (max: 5.6%, min: 0.01) for the analysis of the $O_2$ concentration at the start and at the end of the incubation, respectively.

The original text has been modified to clarify the point raised by the referee about the number of Winkler bottles used in each treatment (Line 188).

- 15. L. 249. "to compare non-parametric paired samples" is an odd phrasing.

The sentence has been deleted from the Statistical analysis section.

- 16. L. 293. It does not seem so obvious to me.

The sentence has been rephrased to improve clarity (Lines 268-270).

- 17. L. 302 and L. 310. Why using RMANOVA for comparing differences at one single sampling point?. I do not follow the rationale for using the two statistical tests here. There is some confusion about statistics throughout the manuscript. The authors should clearly state which tests they used and why or try a different analysis (see my general comment) with changes integrated over the course of the incubation of phase 2.

The statistical analysis has been revised throughout the whole manuscript. A new approach using a two-way ANOVA statistical test has been done in the present version of the manuscript. A detailed explanation of this topic has been included in the Statistical analysis section (Lines 222-229).

- 18. L. 328. "biased" is probably not the best word here.

The term "biased" has been replaced by the more appropriate term (Line 310).

- 19. L. 329. Replace "on" by "of".

**The correction has been done as indicated.**

- 20. L. 341. "there have been…simulate" reads awkward. Please rephrase.

**The sentence has been rephrased to improve clarity as suggested by the referee (Line 327).**

- 21. L. 345. Are the authors sure of this statement?

**The original statement has been modified in accordance with the information from the cited study (Line 330).**

- 22. L. 370-371. This is not true in view of the different sampling points and the data shown in the corresponding figures.

**The new approach proposed by the referee using the integral of the cumulative responses along the incubation (see answer to general comment 1) helps to obtain a global response of the obtained results and reinforce the author's findings from the previous version. The sentence has been corrected taking into account the referee´s comment (Line 377)**

- 23. L. 396-397. "having…production rates" is not correct English usage.

**The sentence has been corrected (Line 402).**

- 24. L. 398-400. Please see my previous comment about lability.

**Please, see answer to specific comment above (Question #2).**

- 25. L. 432. Do the authors imply that their water samples collected on June 27th were "cold"? Maybe there was a strong upwelling on that day but this information is not provided and ca. 15 C is not exactly cold.

**No, we were not referring to the temperature of the natural waters but to the possibility of having active extracellular enzymes coming from the first incubation after freezing the sample, as happened in the reference cited in the manuscript (i.e. Steen & Arnosti 2011). The sentence has been modified for improving the understanding on this issue (Line 452).**

- 26. L. 454. Respiration rate and organic matter are not independent.

**The text has been modified taking into account the referee´s point (Lines 470-472).**

- 27. L. 455. These results are far from "demonstrating" that claim.

**The sentence has been deleted.**

- 28. I am not conviced that "acidified organic matter" and "non acidified organic matter" are the best terms for their treatments, did they check that the resultant DOM was of lower pH in the former treatment? Fig. 5B simply shows that the sample water had a lower pH but not that the DOM was indeed of lower pH.

**The text has been changed throughout all the manuscript to clarify that the pH was measured in the water sample that contains the organic matter and to avoid possible misunderstandings related to the DOM pH. We have also explained in the new text that the treatment tittles (i.e. "acidified organic matter" and "non acidified organic matter") are only a brief way of naming the treatments using the key words for each experimental factor, but they do not imply that the DOM was acidic after bubbling the sample with 1000 ppm $CO_2$. (Lines 140-144)**

- 29. μ M is not the appropriate SI unit, please correct it to _mol L-1.

**Axis nomenclature has been corrected in the text and in figures 3-B and 6.**

- 30. Fig. 4. Please replace the "P" in the Y-axes by TOC, POC and DOC. This is not a very relevant figure and can be eliminated or moved to supplementary information.

**Axis nomenclature has been corrected in figure 4. Following the referee´s suggestion the figure has been moved to supplementary material (Appendix 1).**

- 31. The authors use total abundance of bacteria but probably data about the contribution of low and high nucleic acid content (LNA and HNA) cells are available, as well as some indication of changes in cell size that would provide a good estimate of bacterial biomass that could be compared with changes in BP, even if they were assuming data from the literature to convert from leucine incorporation rates to carbon units.

**Unfortunately we are not able to provide the information requested by the referee since Dr. Luis Lubian, that was the coauthor in charge of the flow cytometry results, passed away recently and did not leave any data related to the information requested by the referee.**

- 32. Dubbing cells able to reduce CTC as "viable" is not the best term. Most authors, including the cited references, refer to them as cells actively respiring or showing active respiration but the number of viable cells in their incubations was likely much higher, just by comparing the cell abundance numbers of Figs. 6 and 8. It is uncommon to show CTC positive cells before total bacterial abundance. Also, why not showing the dynamics of CTC positive bacteria for the entire experiment rather than only at day 7?

**The term "viable", when naming the cells that were able to reduce CTC, has been replaced by cells that were actively respiring, as proposed by the referee.**

**The reason why the CTC positive bacteria test was only performed at day 7 is that this analysis was not included in the preliminary experimental planning. It was proposed after seeing the significant respiration responses at the beginning of the incubation.**

**Regarding the reason for showing the CTC results before the bacterial abundance is that they were more closely related to the respiration rates, which at the same time, were related to the pH and $CO_2$ responses, normally shown at the beginning of the Results section.**

- 33. Fig. 9B. BGE is given either as a percentage or as a ratio, what does "r.u." mean? Also, given the use of a very high and constant leucine-to-carbon conversion factor of 3.1 kg C mol Leu-1, if the values are close to 80% BGE, they are extremely high, very difficult to reconcile with almost no net uptake of DOC (Fig. 7).

**The units in the axis titles have been corrected in both, figure 9-A and 9-B.**

**Since no net uptake can be quantified based on the experimental design of our study (please see answer to question #2) we cannot really correlate DOM results to BGE. BGE values shown in our study are similar and within the range of the values obtained from previous studies carried out with samples of the Ría de Vigo with additions of organic matter (Martínez-García et al. 2010).**

**A NEW VERSION OF THE MANUSCRIPT, INCLUDING ALL THE SUGGESTIONS RAISED BY THE REVIEWER, IS INCLUDED AT THE END OF THE REFEREE´S RESPONSE AS A POSSIBLE NEW VERSION OF THE MANUSCRIPT.**

*Correspondence to*: Cristina Sobrino. +34 986 818789, sobrinoc@uvigo.es

Running head: Ocean acidification and DOM on bacteria

**Abstract.** Microcosm experiments to assess bacterioplankton response to phytoplankton-derived organic matter obtained under current and future-ocean $CO_2$ levels were performed. Surface seawater enriched with inorganic nutrients was bubbled for 8 days with air (current $CO_2$ scenario) or with a 1000 ppm $CO_2$–air mixture (future $CO_2$ scenario) under solar radiation. The organic matter produced under the current and future $CO_2$ scenarios was subsequently used as inoculum. Triplicate 12 L flasks filled

with 1.2 µm-filtered natural seawater enriched with the organic matter inocula were incubated in the dark for 8 days under $CO_2$ conditions simulating current and future $CO_2$ scenarios to study the bacterial response. The acidification of the media increased bacterial respiration at the beginning of the experiment while the addition of the organic matter produced under future levels of $CO_2$ was related to changes in bacterial production and abundance. This resulted in 67% increase in the integrated bacterial respiration under future $CO_2$ conditions compared to present $CO_2$ conditions and 41% higher integrated bacterial abundance with the addition of the acidified organic matter compared to samples with the addition of non acidified organic matter. This study demonstrates that the increase in atmospheric $CO_2$ levels can affect bacterioplankton directly by changes in the respiration rate and indirectly by changes on the organic matter with concomitant effects on bacterial production and abundance.

KEY WORDS: bacterioplankton, phytoplankton, organic matter, ocean acidification.

**1 Introduction**:

The increase in fossil fuel burning, cement production and deforestation together with changes in land use have resulted in an accumulation of atmospheric $CO_2$ at levels never seen before the last two million years (Caldeira & Wickettt 2008, Le Quere et al. 2015). Atmospheric gases can freely diffuse into the ocean surface, which has already absorbed about 30% of the emitted anthropogenic $CO_2$, perturbing the carbonate system and decreasing ocean pH in a process known as ocean acidification (Sabine et al. 2004, Burke et al. 2014).

The latest IPCC report shows that the pH of surface ocean waters has already decreased by 0.1, corresponding to a 26% increase in acidity. If mitigation strategies for global change are not adopted and $CO_2$ emissions continue as usual, ocean pH values will drop about 0.3-0.7 units by the end of the 21$^{st}$ century (Burke et al. 2014). The decrease in seawater pH has strong effects on the ecosystem, the aquatic organisms and the interactions among them. Studies about its consequences in the surface ocean have been primarily focused on calcifying organisms such as corals or coralline algae because they participate in the formation of habitats and human services (Langdon et al. 2003, Fabry et al. 2008). Recent meta-analysis studies also revealed decreased survival, growth, development and abundance of a broad range of marine organisms, although the magnitude of these responses varies between taxonomic groups, including variation within similar species (Kroeker et al. 2013). Additionally, other authors have demonstrated that ocean acidification can increase growth, primary production and $N_2$ fixation rates in some phytoplankton species (Barcelos e Ramos et al. 2007, Fu et al. 2007, Levitan et al. 2007, Iglesias-Rodriguez et al. 2008). In contrast with the abundant information about phytoplankton, very little is known about the response of bacterioplankton.

Heterotrophic bacteria play an important role in the planktonic community since they are responsible for the majority of the organic matter remineralization (Cole et al. 1988, Azam et al. 1998, Nagata et al.

65    2000) allowing the primary producers to make use of the recycled inorganic nutrients. They also return dissolved organic carbon (DOC) to the marine food web via its incorporation into bacterial biomass through what it is called the microbial loop (Azam et al. 1983). However, the microbial response can change depending on phytoplankton taxonomic composition and the nutrient levels, and therefore productivity, of the water (Teira et al. 2012, Bunse et al. 2016, Sala et al. 2016). Despite this important

70    role of bacterioplankton in the marine food web and biogeochemical cycles, only few studies have been designed to elucidate the effects of ocean acidification on bacteria metabolism or its interaction with the abiotic (i.e. temperature, ultraviolet radiation, mixing, etc.) and biotic factors (i.e. algal derived organic matter, microbial community composition, trophic interactions, etc.). Some of these studies suggest an absence of significant metabolic responses in experiments where $CO_2$ levels were manipulated

75    (Rochelle-Newall et al. 2004, Allgaier et al. 2008, Newbold et al. 2012). Fast acclimation of the metabolic machinery to low pH values might have occurred in those experiments 
[revised manuscript text omitted]
 the organic matter produced under current $CO_2$ conditions in the previous incubation (from herein named as Non acidified Organic matter, NO, n=6) and the other half with the organic matter produced under future $CO_2$ conditions by acidifying the media as explained above (from herein named as Acidified Organic matter, AO, n=6). In each case, three replicates were aerated with ambient $CO_2$ levels (Low Carbon treatment (LC): LC_NO, LC_AO) or air with 1000 ppm $CO_2$ (High Carbon treatment (HC): HC_NO, HC_AO). This experimental setup produced 4 different treatments from the less modified sample (LC_NO) to the most altered sample (HC_AO) (Fig. 2). The bottles were located in a walk-in growth chamber under dark conditions at 15 ºC, similar to the *in situ* temperature. The dark incubation for the bacterioplankton samples was aimed to focus the experiment on the two factors of the study, $CO_2$ and organic matter addition, avoiding potential effects of solar radiation on bacterioplankton and organic matter that might complicate the interpretation of the results.

*2.2 DIC, pH and $CO_2$ analysis*

[revised manuscript text omitted]

*2.7 Statistical analysis*

Data followed a normal distribution and homoscedasticity, tested by Jarque-Bera test and Bartlett test, respectively. Thus, a parametric one-way ANOVA or two-way ANOVA statistical tests were used to determine differences between one or two experimental factors, respectively (i.e. $CO_2$ in the first incubation, $CO_2$ and organic matter addition in the second incubation). If the interaction between the two factors was significant a multiple comparison post-test (MCP-test) was carried out. The confidence level was established at the 95%. Statistical analysis was performed using the software package MatLab R2012b.

**3 Results:**

During the first incubation, aimed to obtain the organic matter inocula under current and future $CO_2$ conditions, the LC treatment $pCO_2$ values were close to the atmospheric equilibrium, with values ranging between $419 \pm 21$ ppm $CO_2$ on day 0 and $226 \pm 38$ ppm $CO_2$ on day 3 (mean and SD, n=3) (Fig. 3A). In the HC treatment $pCO_2$ values increased since the beginning of the incubation until reaching values around 1200 ppm the last four days. Maximum values were observed at day 5 with $1227 \pm 149$ ppm $CO_2$. Chl *a* used as an indicator of the phytoplankton biomass showed similar trends in the two treatments. It increased with the initial addition of inorganic nutrients showing an early bloom

on day 1 with Chl *a* values of 21 ± 4 µg L$^{-1}$ and 22 ± 9 µg L$^{-1}$ for LC and HC treatments, respectively.

320 Chl *a* concentration decreased after the bloom, keeping values close to the lowest concentrations on day 6 for the HC treatment with 3.0 ± 0.4 µg L$^{-1}$ and on day 5 for the LC treatment, 3.1 ± 0.5 µg L$^{-1}$ (Fig. 3A). Analysis of phytoplankton composition by flow cytometry did not show significant differences in community structure between LC and HC treatments (data not shown).

Primary production rates followed the Chl *a* pattern with a marked peak blooming the first incubation
325 day followed by a decline afterwards in all the treatments (Figure 1 in Appendix 1). The increase in total carbon fixation during the bloom was due to an increase in both, POC and DOCp production, but it was mainly due to POC assimilation. The percentage of extracellular release of dissolved carbon (PER=DOCp/(POC+DOCp)) ranged between 18% and 77%. DOCp, POC, and therefore TOC, were higher during the bloom under the LC treatment but not significantly different than the rates observed in
330 the HC treatment (1-way ANOVA, TOC: p=0.307, POC: p=0.242, DOCp: p=0.527, n=6). Differences in the production rates between both treatments became negligible after the second incubation day (Figure 1 in Appendix 1).

Bulk DOC concentration increased from day 0 to maximum values on day 7, and similar to Chl *a* concentration and production rates, there were not significant differences between the two $CO_2$
335 treatments at the end of the incubation (1-way ANOVA, p=0.096, n=6) (Fig. 3B). Parallel analysis of DOM fluorescence (*i.e.* protein-like and humic-like substances) also supported the later results (data not shown).

In the second incubation, aimed to assess the effects of $CO_2$ and the organic matter additions on bacterioplankton, $pCO_2$ and pH were similar within the same $CO_2$ treatment (*i.e.* LC_NO and LC_AO

*vs.* HC_NO and HC_AO), but significantly different between LC and HC treatments (Fig. 4A and B).

355 $pCO_2$ in the LC ranged between $350 \pm 28$ ppm and $568 \pm 187$ ppm $CO_2$ (mean and SD, n=3) on day 5 and 2, respectively. In contrast, HC treatments increased from $397 \pm 18$ ppm on day 0 to a maximum value of $2213 \pm 229$ ppm on day 3, significantly different that the expected 1000 ppm $pCO_2$ in the HC treatments. The maximum was followed by a pronounced decrease on day 4, and subsequently, the values were similar to the bubbled air concentrations ($1011 \pm 75$ ppm on day 5) (Fig. 4A). As expected

360 from the $pCO_2$, pH values in the LC treatments were fairly constant with a mean value of 8.07 but decreased markedly from 8.03 on day 0 to 7.51 on day 3 in the HC treatment. After this minimum, pH increased to values around 7.8 until the end of the experiment (Fig. 4B).

The changes in $CO_2$ concentration and pH were accompanied with an increase in BR in the HC treatments (Fig. 4C). BR was also fairly constant in the LC treatments, but showed a pronounced

365 increase from day 0 to day 2 being significantly different in the HC treatments (2-way ANOVA, p=0.048, n=12). After reaching the maximum, bacterial respiration dropped to similar values than those observed in the LC treatments (Fig. 4C). Among the LC treatments, the LC_NO treatment showed the lowest variability in the respiration rates and the highest values at the end of the incubation (1-way ANOVA and MCP-test, p<0.001, n=6). Flow cytometry analysis of the CTC-positive bacteria, which

370 have been previously related to actively respiring bacteria, also showed that the LC_NO treatment had significantly higher values than the other treatments on day 7 (1-way ANOVA and MCP-test, p=0.007, n=6). In contrast, the other three treatments did not show significant differences among them (Fig. 5). Similar to $pCO_2$ and pH, statistical differences between samples with organic matter inocula produced under current and future $CO_2$ scenarios within each $CO_2$ treatment were not significant

regarding respiration rates. The analysis of the cumulative responses, measured as the integrated bacterial respiration during the whole incubation, showed 67% higher values under HC than under LC conditions (2-way ANOVA, p=0.010, n=6) (Fig. 4D).

The initial concentration of dissolved organic carbon (DOC, $\mu mol\ C\ L^{-1}$) in the samples was quite similar among the four treatments until the peak in respiration (Fig. 6). DOC before the addition of the organic matter inocula was 89 $\mu mol\ C\ L^{-1}$ and increased in all the treatments to maximum values of 111 ± 5 $\mu mol\ C\ L^{-1}$ and 117 ± n.d. $\mu mol\ C\ L^{-1}$ in treatments LC_AO and HC_NO on day 3, respectively. Afterwards DOC in the LC treatments kept approximately constant but decreased 27% in the HC treatments compared to the LC treatments (81 ± 3 $\mu mol\ C\ L^{-1}$ and 81 ± 2 $\mu mol\ C\ L^{-1}$ in HC_NO and HC_AO, respectively). Statistical analysis showed significant differences between both $pCO_2$ treatments on day 7 (2-way ANOVA, p<0.001, n=12) (Fig. 6).

Unlike $pCO_2$, pH, respiration and DOC, bacterial abundance and production showed differences regarding the origin of the organic matter. The addition of the organic matter inocula produced a fast increase in production and abundance from day 0 to day 3 in all the treatments. Cell abundance increased from 6.2 ± 0.4 x $10^4$ bacteria $mL^{-1}$ before the addition of the organic matter inocula to a maximum of 9.0 ± 0.4 x $10^5$ bacteria $mL^1$ in the HC_AO treatment (Fig. 7A). Differences for the bacterial abundance started to be significant in day 1 (1-way ANOVA and MCP-test, p=0.012, n=6) but they were more clearly observed during the maximum at day 3 (1-way ANOVA and MCP-test, p<0.001, n=6). Bacterial abundances were 29% and 31% higher in samples where the acidified organic matter was added than in those with the addition of organic matter produced under the current $CO_2$ scenario, in the LC and HC treatments, respectively (Fig. 7A). This resulted in 41% higher integrated

bacterial abundance with the addition of the acidified organic matter than in samples with the addition of non acidified organic matter (2-way ANOVA, p<0.001, n=12) (Fig. 7B).

Additionally, bacterial production increased from a minimum value of $1.1 \pm 0.1$ µg C $L^{-1}$ $d^{-1}$ on day 0 to maximum values on day 2 for the four treatments, ranging between $185 \pm 37$ µg C $L^{-1}$ $d^{-1}$ and $208 \pm 4$ µg C $L^{-1}$ $d^{-1}$ for HC_NO and LC_NO, respectively. However, the analysis of the integrated response did not show significant differences among treatments (Fig 7.D). Only punctually, the treatments with the addition of acidified organic matter (LC_AO and HC_AO) showed a higher decrease in the production rates than those with the addition of the non-acidified organic matter, resulting in significant differences later on (2-way ANOVA, p=0.001, n=12). On day 7 treatments LC_NO and HC_NO produced 53% and 45% more than treatments LC_AO and HC_AO, respectively (Fig. 7C).

The BCD was affected by the respiratory activity at the beginning of the incubation (day 1, 2-way ANOVA, p=0.041, n=12) and by the production at the end (day 7, 2-way ANOVA, p=0.002, n=12) (Fig 8A). In consequence, the BGE was higher for the LC treatments, at the beginning of the incubation during the activity peak, but decreased at the end. At day 7 the HC treatment with the addition of non-acidified organic matter (HC_NO) showed significantly higher efficiency than the other treatments (1-way ANOVA and MCP-test, p=0.002, n=6) (Fig 8C). Overall, integrated BCD resulted 24% higher under HC conditions while the BGE was 11% lower in the same conditions, compared to LC conditions (2-way ANOVA, p=0.011, p=0.019, respectively, n=6) (Fig 8B, D).

**4 Discussion:**

The main goal of the current study was to distinguish between the direct and indirect effects of ocean acidification on natural bacterial assemblages. To achieve this objective we performed a 2×2 experimental design combining the acidification of seawater and the addition of phytoplankton-derived organic matter produced under current and future $CO_2$ conditions and natural solar exposures. UVR induces photomineralization of coloured dissolved organic matter (CDOM) increasing the biological availability of the resulting DOM (Moran & Zepp 1997) and can also increase DOCp production in surface waters (Carrillo et al. 2002, Helbling et al. 2013, Fuentes-Lema et al. 2015).

Although there have been described different methodologies to modify the seawater pH to simulate an ocean acidification scenario, the continuous bubbling of the natural plankton communities with a target $CO_2$ concentration of 1000 ppm $CO_2$ was chosen in the present study to simulate the $pCO_2$ and pH conditions expected for the end of the century. This method simulates the natural variations of sea surface $pCO_2$ driven by differences between atmospheric and sea water $CO_2$ concentration providing quite realistic responses of the organisms to this environmental factor (Rost et al. 2008). It also allows that changes in the biological activity of the samples enable the modification of the $pCO_2$ values if, for example, the photosynthetic or respiratory rates become faster than the rates needed to achieve the $CO_2$ chemical equilibrium in seawater. Changes in $pCO_2$ values due to microbial activity are usually observed in natural waters during bloom events in surface waters or in areas with high amount of organic matter (Joint et al. 2011). $pCO_2$ also increases with depth due to the increase in heterotrophic activity compared to the autotrophic activity in surface (Pukate & Rim-Rukeh 2008, Dore et al. 2009) and can change due to upwelling events, for example in the same area where the samples were

collected, reaching values close to those observed in the present work (Alvarez et al. 1999, Gago et al. 2003).

In our study, the $CO_2$ enrichment did not produce a significant effect on phytoplankton production or biomass, measured as $^{14}C$ incorporation or Chl *a* concentration, respectively. Phytoplankton community composition changed from bigger to smaller phytoplankton cells, as has been observed in similar microcosm studies (Reul et al. 2014, Grear et al. 2017), but differences between present and future $CO_2$ treatments were not observed. DOCp production increased during the bloom evolved at the beginning of the incubation but bulk DOC concentration showed similar values between $CO_2$ treatments, as expected based on the lack of differences observed in the biological and metabolic variables. Despite several studies indicate an increase in phytoplankton carbon production and biomass under future scenarios of $CO_2$ (Kroeker et al. 2013), in our study exposure of the cells to natural conditions including solar UVR might have counteracted the stimulatory effect of the high $CO_2$, since it increases the sensitivity of phytoplankton to photoinhibition (Sobrino et al. 2008, 2009, Gao et al. 2012).

The addition of the organic matter and the start of the aeration in the second incubation produced a burst in the metabolic activity of the bacteria. However this increase in bacterial activity was not followed by a significant decrease in DOC. Taking into account the experimental design of the study keeping both, the particulate and the dissolved fractions of the organic matter from the first phase incubation, it is likely that production of DOC exceeded its consumption by heterotrophic bacteria, leading to a net accumulation of DOC in the microcosms. The particulate fraction could potentially have been degraded during the second phase of the experiment due to the elevated $CO_2$ concentrations or the enzymatic activity (Piontek et al. 2010). Enzymatic activity on POC depends on the type of substrate and relates to

the metabolic capabilities of the microbial community but can act on the temporal interval of our experiment (Arnosti et al. 2005). Counterintuitively, the high metabolic activity observed after adding the organic matter might have also contributed to the net accumulation of DOC since it has been recently demonstrated that microheterotrophs can release part of the substrate as new DOM (Lønborg et al. 2009). Freezing the sample with the organic matter produced in the first incubation might have also contributed to the DOC production during the second incubation. Freezing was necessary to avoid an excessive consumption of the acidified and non acidified organic matter for later use, since DOC released by marine phytoplankton is highly labile to bacterial utilization and can be degraded significantly within hours (Chen & Wangersky 1996). However, results from Peacock et al. (2015) suggest that changes in DOC concentrations by freeze/thaw cycles in samples stored no longer than six months are less than 5% and do not show a clear pattern of increase or decrease (Peacock et al. 2015). Such changes are within the range of typical precision for DOC analysis and should not be significantly different among the experimental treatments since all the samples were processed in a similar way.

Bacterioplankton growing in the HC treatments showed higher rates of respiration the first two days. This response was opposite to that described for some bacterial cultures (Teira et al. 2012) and seems to be related to an acclimation to the new pH values, somehow similar to observed for phytoplankton (Sobrino et al. 2008). Consequently, $p\text{CO}_2$ in the HC treatments increased more than expected due to the biological activity carried out by bacterioplankton. The increase in respiration was also parallel to an increase in bacterial abundance and production. However, the results indicate that while changes in respiration were related to $p\text{CO}_2$ values, changes in bacterial abundance were mainly related to the origin of the organic matter amendments. The bacterial abundance was stimulated by the presence of

[revised manuscript text omitted]

the high metabolic activity observed after adding the organic matter might have also contributed to the net accumulation of DOC since it has been recently demonstrated that microheterotrophs can release part of the substrate as new DOM (Lønborg et al. 2009).¶
¶
most bacteria maintain an intracellular pH between 7.4 to 7.8 (Padan et al. 2005), which might result in energy savings when extracellular pH approaches to the predicted values for future scenarios of ocean acidification. Additionally, results from bacteria cultures evidenced that the response to elevated CO2 depends on the species, stimulating the high CO2 concentrations the growth efficiency in Flavobacteria but without significant effects on Rhodobacteria (Teira et al. 2012).¶
¶

610 further studies are needed to disentangle the lack of agreement among bacterial production, abundance and growth efficiency, more specially during the peaks of activity since they seem to show the biggest uncoupling between the measured parameters (del Giorgio & Cole 1998). Lack of total agreement regarding the statistical analysis might be related to differences in methodological sensitivity and variability. Moreover, leucine (or thymidine) to carbon ratios can change dramatically in different

615 environmental conditions (Teira et al. 2015) and bacteria cultures under controlled conditions in the lab have shown that $CO_2$ fixation can increase under high carbon conditions, representing 8 to 9% of the bacterial production (Teira et al. 2012). All these factors can affect the bacterial production calculation. Despite our work used a conversion factor that had being empirically tested for natural samples collected from the same area without significant differences between stations with different $CO_2$

620 concentrations (Martínez-García et al. 2010), other factors such as the bacterial composition or the nature of organic matter additions might be affecting the final production rates. An attempt to empirically estimate the leucine-to-carbon conversion factors should be addressed in order to decrease uncertainties related to the bacterial production and the BCD and finally enhance our understanding in this topic.

625 In summary, the results from this investigation show that ocean acidification can significantly affect bacterioplankton metabolism directly by changes in the respiration rate and indirectly on bacterial abundance by changes in the organic matter.

>

*Acknowledgements:* This study was possible thanks to the financial support from Xunta de Galicia

630 (7MMA013103PR project and ED431G/06 Galician Singular Research Center) and Spanish Ministry of

Economy, Industry and Competitiveness (CTM2014-59345-R project and H. Sanleón-Bartolomé fellowship (FPI-IEO fellowship 2011/06)), and the technical assistance from Maria José Fernández Pazó, Vanesa Vieitiez and ECIMAT staff. The authors are especially grateful to Xosé A. Alvarez-Salgado and Marta Álvarez for their scientific support and suggestions on the manuscript and to María Pérez Lorenzo and Marta Ruiz Hernández for their help with the respiration and bacterial production techniques, respectively. The authors of this study do not have conflict of interest to declare.

640

**6 Figure legends:**

Figure 1. Geographical location of Ría de Vigo in the NW Iberian Peninsula. The insert shows a more detailed map of the Ría de Vigo and the locations of A) ECIMAT and B) sampling station.

Figure 2. Experimental design of the study.

Figure 3. A) Phytoplankton biomass measured as Chl $a$ concentration and $pCO_2$ evolution along the first incubation period aimed to obtain the organic matter inocula under current and future $CO_2$ conditions. Black and striped vertical bars correspond to the Chl $a$ mean $\pm$ SD (n=3) ($\mu$g L$^{-1}$) obtained under high and low carbon (HC and LC) treatments, respectively. Black and grey circles correspond to the $pCO_2$ mean $\pm$ SD (n=3) (ppm $CO_2$) in HC and LC treatments, respectively. B) Temporal evolution of the dissolved organic carbon (DOC) concentration ($\mu$mol C L$^{-1}$)) during this first incubation. The black and grey dots indicate the mean $\pm$ SD (n=3) of DOC from HC and LC treatments, respectively.

Figure 4. A) $pCO_2$ (ppm $CO_2$) and B) pH values in the four treatments of the second incubation period, respectively. Mean $\pm$ SD (n=3). C) Temporal evolution of bacterial respiration ($\mu$mol $O_2$ d$^{-1}$) and D) cumulative bacterial respiration ($\mu$mol $O_2$) during the second incubation period in the four treatments, mean $\pm$ SD (n=3). In all figures, black and grey triangles represent HC_NO and HC_AO treatments, respectively. Black and grey circles correspond to LC_NO and LC_AO treatments. The asterisk indicate

significant differences between LC *vs*. HC treatments with p-value < 0.05. The three asterisks indicate a significant difference between LC_NO with respect to the other treatments with p-value < 0.001.

860

Figure 5. Bacterial viability (Active bacterial mL$^{-1}$) measured with the CTC dye on day 7, mean ± SD (n=3). The two asterisks indicate mean significant difference with the other treatments with p-value < 0.01.

865 Figure 6. Temporal evolution of the dissolved organic carbon (DOC) concentration (µmol C L$^{-1}$) during the second incubation period, mean ± SD (n=3). Black and grey triangles represent HC_NO and HC_AO treatments, respectively. Black and grey circles correspond to LC_NO and LC_AO treatments, respectively. The three asterisks indicate significant differences between LC *vs*. HC treatments with p-value < 0.001.

870

Figure 7. A) Histogram of bacterial abundance evolution (Cells mL$^{-1}$), treatments with different letter indicate significant differences (p-value < 0.01), and B) cumulative response of bacterial abundance (Cells mL$^{-1}$) in the four treatments during the course of the second incubation period, mean ± SD (n=3).C) Bacterial production (µg C L$^{-1}$ d$^{-1}$) and D) cumulative bacterial production (µg C L$^{-1}$) from the 875 four treatments during the second incubation period, mean ± SD (n=3). The two asterisks indicate significant differences between NO *vs*. AO treatments with p-value < 0.01. Black and grey triangles represent HC_NO and HC_AO treatments, respectively. Black and grey circles correspond to LC_NO and LC_AO treatments, respectively.

880 Figure 8. Temporal evolution of A) bacterial carbon demand (BCD ($\mu$mol C L$^{-1}$ d$^{-1}$)), B) cumulative

BCD ($\mu$mol C L$^{-1}$), C) bacterial growth efficiency (BGE) and D) cumulative BGE during the second

incubation calculated from the bacterial production and respiration rates obtained from this study. One

asterisk indicate significant differences with p-value $< 0.05$ between HC *vs.* LC treatments. Two

asterisks indicate significant differences with p-value $< 0.01$ between NO *vs* AO treatments, for the

885 BCD and between HC_NO *vs.* the rest of the treatments (LC_NO, LC_AO and HC_AO) for the BGE.

Black and grey triangles represent HC_NO and HC_AO treatments, respectively. Black and grey circles

correspond to LC_NO and LC_AO treatments, respectively.

FIGURE 1:

[Figure]

FIGURE 2:

[Figure]

FIGURE 3:

[Figure]

FIGURE 4:

[Figure]

FIGURE 5:

[Figure]

895    FIGURE 6:

[Figure]

[Figure]

[Figure]

**7. Appendix 1:**

*Primary production*

Incubations were performed at noon and lasted 3 to 3.5 hours. Fifteen mL samples of each microcosm were inoculated with $H^{14}CO_3^-$ (approximately 1 µCi mL$^{-1}$ final concentration) and incubated in UVR transparent Teflon-FEP bottles under full solar radiation exposures in a refrigerated tank contiguous to the experimental microcosms. The teflon bottles were tied on top of a UVR transparent acrylic tray, keeping all bottles under flat and constant position. Tray was wrapped with 2 layers of neutral density screen to obtain saturating but non-photoinhibitory solar exposures. For analysis of the fraction of the fixed carbon incorporated into particulate (POC) and dissolved (DOCp) organic carbon, 5 mL samples were filtered through 0.2 µm PC filters (25 mm diameter) under low pressure (50 mm Hg) after the light incubation period, using 2 manifolds simultaneously (10 positions per manifold). POC was retained on the filter while the filtrate was directly collected in scintillation vials to assess $^{14}C$ activity in the dissolved fraction (DOCp). Simultaneously, the total amount of organic carbon incorporated in the cells (TOC) was measured independently of the DOCp-POC filtration by processing 5 mL of the incubated samples. Non-assimilated $^{14}C$ was released by exposing the filters (POC) to acid fumes (50% HCl) or by adding 200 µl of 10% HCl to the liquid samples (DOCp & TOC, respectively) and shaking overnight. The radioactivity of each sample was measured using a Wallac WinSpectral 1414 scintillation counter (EG&G Company, Finland). Data analysis determined that both, TOC and POC+DOCp results, were significantly correlated (y= 1.05x (±0.07) + 0.05 (±0.03), $R^2$=0.85, ANOVA, p=0.637, n=34).

920 Figure 1. Primary production rates measured as the incorporation of $^{14}$C into organic compounds during the first incubation period aimed to obtain the organic matter inocula under current and future $CO_2$ conditions A) Total organic carbon fixation rates obtained from a different sample than the sample used for the determination of POC and DOCp fixation rates (TOC) B) Particulate organic carbon fixation rates (POC) C) Rates of dissolved organic carbon production from phytoplankton origin (DOCp). Mean

925 ± SD (n=3).